# Inserting Anybody in Diffusion Models via Celeb Basis

**Ge Yuan**[1,2,3,4][*]  **Xiaodong Cun**[4]  **Yong Zhang**[4]  **Maomao Li**[4][†]  **Chenyang Qi**[5]
**Xintao Wang**[4]  **Ying Shan**[4]  **Huicheng Zheng**[1,2,3][†]

[1] School of Computer Science and Engineering, Sun Yat-sen University
[2] Key Laboratory of Machine Intelligence and Advanced Computing, Ministry of Education, China
[3] Guangdong Key Laboratory of Information Security Technology
[4] Tencent AI Lab    [5] The Hong Kong University of Science and Technology

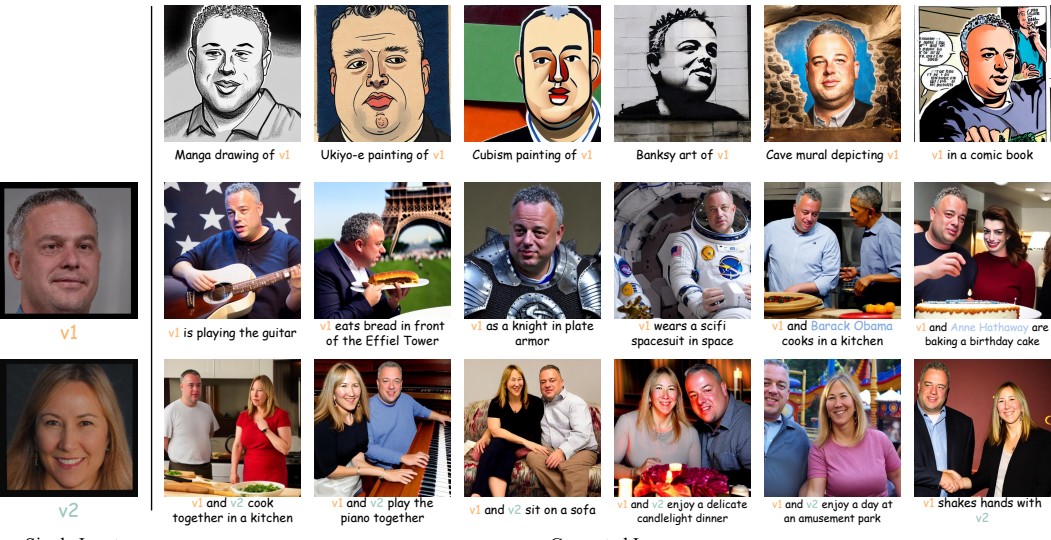

Figure 1: Given a single facial photo ($v1$ or $v2$) as a tunable sample, the proposed method can insert this identity into the trained text-to-image model, *e.g.*, Stable Diffusion [1], where the new person ($v1$) can act like the original concept in the trained model and interact with another newly trained concept ($v2$). Note that the input images are randomly generated from StyleGAN [2].

## Abstract

Exquisite demand exists for customizing the pretrained large text-to-image model, *e.g.* Stable Diffusion, to generate innovative concepts, such as the users themselves. However, the newly-added concept from previous customization methods often shows weaker combination abilities than the original ones even given several images during training. We thus propose a new personalization method that allows for the seamless integration of a unique individual into the pre-trained diffusion model using just *one facial photograph* and only *1024 learnable parameters* under *3 minutes*. So we can effortlessly generate stunning images of this person in any pose or position, interacting with anyone and doing anything imaginable from text prompts. To achieve this, we first analyze and build a well-defined celeb basis from the embedding space of the pre-trained large text encoder. Then, given one facial photo as the target identity, we generate its own embedding by optimizing the

---

[*]This work was done when the first author was an intern at Tencent AI Lab.
[†]Corresponding authors

37th Conference on Neural Information Processing Systems (NeurIPS 2023).

weight of this basis and locking all other parameters. Empowered by the proposed celeb basis, the new identity in our customized model showcases a better concept combination ability than previous personalization methods. Besides, our model can also learn several new identities at once and interact with each other where the previous customization model fails to. Project page is at: http://celeb-basis. github.io. Code is at: https://github.com/ygtxr1997/CelebBasis.

# 1 Introduction

Vast image-text pairs [3] during training and the powerful language encoder [4] enable the text-to-image models [1, 5, 6] to generate diverse and fantastic images from simple text prompts. Though the generated images are exquisite, they may still fail to satisfy the users' demands since some concepts are not easy to be described by the text prompt [7]. For instance, individuals frequently post self-portraits on social media platforms, where the pre-trained text-to-image models struggle to produce satisfactory pictures of them despite receiving comprehensive instructions. This shortcoming makes these models less attractive to general users.

Recent works [8, 9, 10, 7] solve this problem via efficiently tuning the parameters of the model for personalization usage. For example, these techniques insert the new concepts (*e.g.* a specific bag, a dog, or a person) into the model by representing them as rarely-used pseudo-words [10] (or text-embeddings [9, 7, 8, 11]) and finetuning the text-to-image model with a few of these samples [10, 7]. After training, these pseudo-words or embeddings can be used to represent the desired concept and can also perform some combination abilities. However, these methods often struggle to generate the text description-aligned image with the same concept class (*e.g.* the person identities) [8, 10]. For instance, the original Stable Diffusion [1] model can successfully generate the image of the text prompt:" *Barack Obama and Anne Hathaway are shaking hands.*" Nevertheless, in terms of generating two newly-learned individuals, the previous personalized methods [8, 10, 7] fall short of producing the desired identities as depicted in our experiments (Fig. 5).

In this work, we focus on injecting the most specific and widely-existing concept, *i.e.* the human being, into the diffusion model seamlessly. Drawing inspiration from the remarkable 3DMM [12], which ingeniously represents novel faces through a combination of mean and weight values derived from a clearly defined basis, we build a similar basis to the embeddings of the celebrity names in pretrained Stable Diffusion. In this way, are capable of representing any new person in the trained diffusion model via the basis coefficients.

We first collect a bunch of celebrity names from the Internet and filter them by the pre-trained text-to-image models [1]. By doing so, we obtain 691 well-known names and extract the text embedding by the tokenizer of the CLIP [4]. These celeb names corresponds to over 700K samples in LAION datasets [3] (see our appendix for detailed statistics), containing powerful priors. Then, we construct a *celeb basis* via Principal Component Analysis (PCA [13]). To represent a new person with PCA coefficients, we use a pre-trained face recognition network [14] as the feature extractor of the given photo and learn a series of coefficients to re-weight the celeb basis, so that the new face can be recognized by the pre-trained CLIP transformer encoder. During the process, we only use a single facial photo and fix the denoising UNet and the text encoder of Stable Diffusion to avoid overfitting. After training, we only need to store the 1024 coefficients of the celeb basis to represent the newly-added identity since the basis is shared across the model. Though simple, the concept composition abilities [1] of the trained new individual is well-preserved, as we only reweight the text embeddings of the trained CLIP model and freeze the weights in the diffusion process. Remarkably, the proposed method has the ability to produce a strikingly realistic photo of the injected face in any given location and pose. Moreover, it opens up some new possibilities such as learning multiple new individuals simultaneously and facilitating seamless interaction between these newly generated identities as shown in Fig. 1. The contributions of the paper are listed as follows:

- We propose celeb basis, a basis built from the text embedding space of the celebrities' names in the text-to-image model and verify its abilities, such as interpolation.

- Based on the proposed celeb basis, we design a new personalization method for the text-to-image model, which can remember any new person from a single facial photo using only 1024 learnable coefficients.

- Extensive experiments show our personalized method has more stable concept composition abilities than previous works, including generating better identity-preserved images and interacting with new concepts.

## 2  Related Work

**Image Generation and Editing.** Given a huge number of images as the training set, deep generative models target to model the distribution of training data and synthesize new realistic images through sampling. Various techniques have been widely explored, including GAN [15, 2], VAE [16, 17], Autoregressive [6, 18, 19, 20], flow [21, 22]. Recently, diffusion models [23, 24] gain increasing popularity for their stronger abilities of text-to-image generation [5, 6, 1]. Conditioned on the text embedding of pre-trained large language models [4], these diffusion models are iterative optimized using a simple denoising loss. During inference, a new image can be generated from sampled Gaussian noise and a text prompt. Although these diffusion models can synthesize high-fidelity images, they have difficulties in generating less common concepts [25] or controlling the identity of generated objects [7]. Current editing methods are still hard to solve this problem, *e.g.* directly blending the latent of objects [26, 27] to the generated background will show the obvious artifacts and bring difficulties in understanding the scenes correctly [28]. On the other hand, attention-based editing works [29, 30, 31, 32] only change the appearance or motion of local objects, which can not generate diverse new images with the same concept (*e.g.* human and animal identity) from the referred image.

**Model Personalization.** Different from text-driven image editing, tuning the model for the specific unseen concept, *i.e.* personalized model, remembers the new concepts of the reference images and can synthesize totally unseen images of them, *e.g.* appearance in a new environment, interaction with other concepts in the original stable diffusion. For generative adversarial neworks [2, 33], personalization through GAN inversion has been extensively studied. This progress typically involves finetuning of the generator [34, 35], test-time optimization of the latents [36], or a pre-trained encoder [37]. Given the recent diffusion generative model [1, 5], it is straightforward to adopt previous GAN inversion techniques for the personalization of diffusion models. Dreambooth [10] finetunes all weight of the diffusion model on a set of images with the same identity and marks it as the specific token. Meanwhile, another line of works [7, 9, 38] optimizes the text embedding of special tokens (*e.g.* $V^*$) to map the input image while freezing the diffusion model. Later on, several works [8, 39, 40] combine these two strategies for multi-concept interaction and efficient finetuning taking less storage and time. These methods focus on general concepts in the open domain while struggling to generate interactions between fine-grained concepts, *i.e.*human beings with specific identities. Since most of the previous works require the tuning in the test time, training inversion encoders are also proposed to generate textual embedding from a single image in the open domain (*e.g.* UMM [41], ELITE [42], and SuTI [43]), or in human and animal domain (*e.g.* Taming-Encoder [44], Instant-Booth [45], E4T [11]). However, a general human identity-oriented embedding is difficult to be obtained from a naively optimized encoder, and tuning the Stable Diffusion on larger-scale images often causes the concept forget. In contrast, our method focuses on a better representation of identity embedding in the diffusion model (celeb basis in Sec. 3.1), which significantly eases the process of optimization such that we only need 1024 parameters to represent an identity more correctly as in Sec. 3.2 and stronger concept combination abilities, which can converge in only 3 minutes.

**Identity Basis.** Representing the human identity via basis is not new in traditional computer vision tasks. *e.g.* in human face modeling, 3D Morphable Models [12] and its following models [46, 47] scans several humans and represent the shape, expression, identity, and pose as the PCA coefficients [13], so that the new person can be modeled or optimized via the coefficients of the face basis. Similar ideas are also used for face recognition [48], where the faces in the dataset are collected and built on the basis of the decision bound. Inspired by these methods, our approach takes advantage of the learned celebrity names in the pre-trained text-to-image diffusion model, where we build a basis on this celebrity space and generate the new person via a series of learned coefficients.

## 3  Method

Our method aims to introduce a new identity to the pre-trained text-to-image model, *i.e.* Stable Diffusion [1], from a single photo via the optimized coefficients of our self-built celeb basis. So that it can memorize this identity and generate new images of this person in any new pose and interact with other identities via text prompts. To achieve this, we first analyze and build a celeb basis on

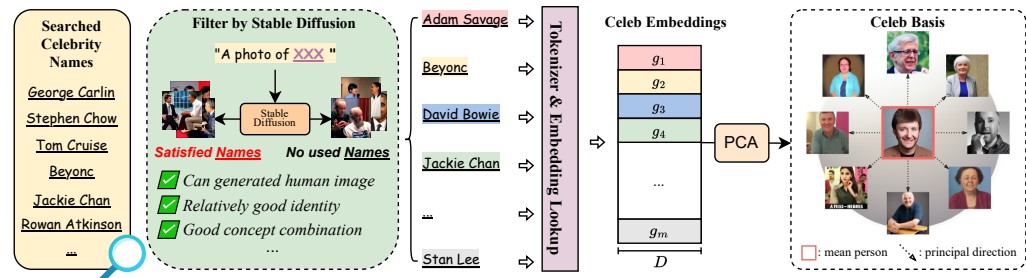

Figure 2: The building process of the proposed Celeb Basis. First, we collect about 1,500 celebrity names as the initial collection. Then, we manually filter the initial one to $m = 691$ names, based on the synthesis quality of text-to-image diffusion model [1] with corresponding name prompt. Later, each filtered name is tokenized and encoded into a celeb embedding group $g_i$. Finally, we conduct Principle Component Analysis to build a compact orthogonal basis, which is visualized on the right.

the embedding space of the text encoder through the names of the celebrities (Sec. 3.1). Then, we design a face encoder-based method to optimize the coefficients of the celeb basis for text-to-image diffusion model customization (Sec. 3.2).

### 3.1 Celeb Basis

**Preliminary: Text Embeddings in Text-to-Image Diffusion Models.** In the text-to-image model, given any text prompts $u$, the tokenizer of typical text encoder model $e_{\text{text}}$, *e.g.* BERT [49] and CLIP [4], divides and encodes $u$ into $l$ integer tokens by order. Correspondingly, by looking up the dictionary, an embedding group $g = [v_1, ..., v_l]$ consisting of $l$ word embeddings can be obtained, where each embedding $v_i \in \mathbb{R}^d$. Then the text transformer $\tau_{\text{text}}$ in $e_{\text{text}}$ encodes $g$ and generates text condition $\tau_{\text{text}}(g)$. The condition $\tau_{\text{text}}(g)$ is fed to the conditional denoising diffusion model $\epsilon_\theta(z_t, t, \tau_{\text{text}}(g))$ and synthesize the output image following an iterative denoising process [23], where $t$ is the timestamp, $z_t$ is a noised image or latent at $t$. Previous text-to-image model personalization methods [7, 10, 9] have shown the importance of text embedding $g$ in personalizing semantic concepts. However, in text-to-image models' personalization, they only consider it as an optimization goal [9, 7, 8], instead of improving its representation.

**Interpolating Abilities of Text Embeddings.** Previous works have shown that text embedding mixups [50] benefit text classification. To verify the interpolation abilities in text-to-image generation, we randomly pick two celebrity names embeddings $v_1$ and $v_2$, and linearly combine them as $\hat{v} = \lambda v_1 + (1 - \lambda)v_2$, where $0 < \lambda < 1$. Interestingly, the generated image of the interpolated embedding $\hat{v}$ also contains a human face as shown in Fig. 3, and all the generated images perform well in acting and interacting with other celebrities. Motivated by the above finding, we build a celeb basis so that each new identity can lie in the space formed by celebrity embeddings.

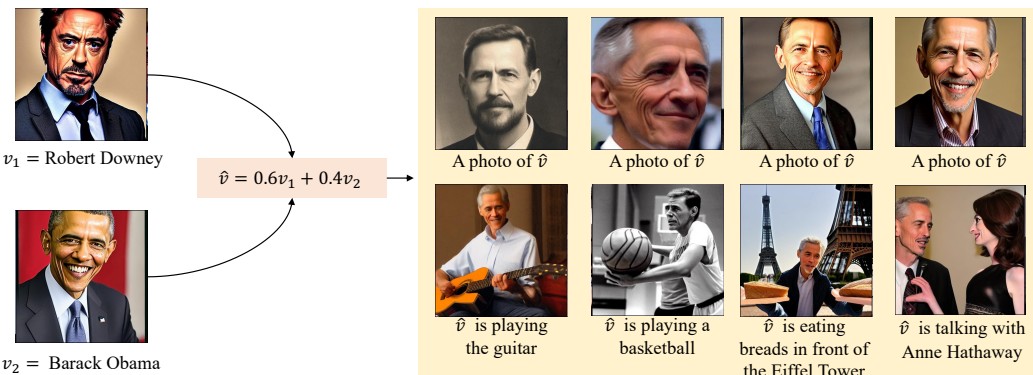

Figure 3: The interpolated text-embedding of two celebrities is still a human (top row) and it also can perform strong concept combination abilities in the pretrained Stable Diffusion [1] (bottom row).

**Build Celeb Basis from Embeddings of the Collected Celebrities.** As shown in Fig. 2, we first crawl about 1,500 celebrity names from Wikipedia as the initial collection. Then, we build a manual filter based on the trained text-to-image diffusion model [1] by constructing the prompts of each name and synthesizing images. A satisfied celeb name should have the ability to generate human images

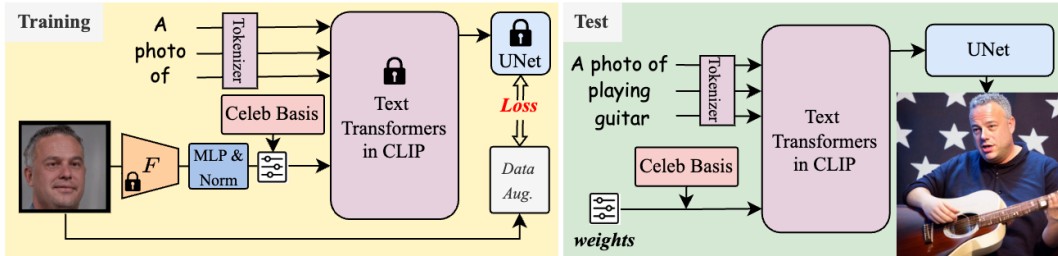

Figure 4: During training (left), we optimize the coefficients of the celeb basis with the help of a fixed face encoder. During inference (right), we combine the learned personalized weights and shared celeb basis to generate images with the input identity.

with prompt-consistent identity and interact with other celebs in synthesized results. Overall, we get $m = 691$ satisfied celeb names where each name $u_i, i \in \{1, ..., m\}$ can be tokenized and encoded into a celeb embedding group $g_i = [v_1^i, ..., v_{k_i}^i]$, notice that the length $k_i$ of each celeb embedding group $g_i$ might not the same since each name may contain multiple words (or the tokenizer will divide the word into sub-words). To simplify the formula, we compose the nonrepetitive embeddings so that each $g_i$ only contains the first two embeddings (*i.e.* $k_i = 2$ for all $m$ celebrities). Using $C_1$ and $C_2$, *i.e.* $C_k = [v_k^1, ..., v_k^m]$, to denote the first and second embeddings of each $g_i$ respectively, we can roughly understand them as the *first name and last name embedding sets*.

To further build a compact search space, inspired by 3DMM [12] which uses PCA [13] to map high-dimensional scanned 3D face coordinates into a compact lower-dimensional space, for each embedding set $C_k$, we calculate its mean $\overline{C}_k = \frac{1}{m} \sum_{i=1}^{m} v_k^i$ and PCA mapping matrix $B_k = \mathrm{PCA}(C_k, p)$, where $\overline{C}_k \in \mathbb{R}^d$ and $\mathrm{PCA}(X, p)$ indicates the PCA operation that reduces the second dimension of matrix $X \in \mathbb{R}^{m \times d}$ into $p$ ($p < d$) principal components, *i.e.* $B_k = [b_k^1, ..., b_k^p]$. As shown in Fig. 2, the mean embedding $\overline{C}_k$ still represents a face and we can get the new face via some coefficients applied to $B_k$.

Overall, our celeb basis is defined on two basis $[\overline{C}_1, B_1]$ and $[\overline{C}_2, B_2]$ working like the first and last name. We use the corresponding principle components $A_1$ and $A_2$ (where $A_k = [\alpha_k^1, ..., \alpha_k^p]$) to represent new identities. Formally, for each new person $\hat{g}$, we use two $p$-dimensional coefficients of the celeb basis and can be written by:

$$\hat{g} = [\hat{v}_1, \hat{v}_2], \quad \hat{v}_k = \overline{C}_k + \sum_{x=1}^{p} \alpha_k^x b_k^x, \tag{1}$$

In practice, $p$ equals 512 as discussed in the ablation experiments.

To control the generated identities, we optimize the coefficients with the help of a face encoder as the personalization method. We introduce it in the below section.

### 3.2 Stable Diffusion Personalization via Celeb Basis

**Fast Coefficients Optimization for Specific Identity.** Given a single facial photo, we use the proposed celeb basis to embed the given face image $x$ of the target identity into the pretrained text-to-image diffusion model as shown in Fig. 4. Since direct optimization is hard to find the optimized weight, we consider using the pre-trained state-of-the-art face recognition models $F$, *i.e.* ArcFace [14], to capture the identity-discriminative patterns. In detail, we adopt the $F$ to extract 512 dimension face embedding as priors. Then a single-layer MLP followed by an $L_2$-normalization is used to map the face priors into the modulating coefficients $A_1$ and $A_2$. Following the Eq. 1, we can obtain the embedding group $\hat{g}$ of the $x$ using the pre-defined basis. By representing the text prompt of $\hat{g}$ as $V^*$, we can involve $V^*$ to build the training pairs between the text prompt of input face and "*A photo of* $V^*$", "*A depiction of* $V^*$", *etc.*. Similar to previous works [10, 7, 8], we only use simple diffusion denoising loss [23]:

$$\mathbb{E}_{\epsilon \sim N(0,1), x, t, g}[\|\epsilon - \epsilon_\theta(z_t, t, \tau_{\text{text}}(g))\|], \tag{2}$$

where $\epsilon$ is the unscaled noise sample, $g$ denotes the text embeddings containing $\hat{g}$. During training, only the weights of MLP need to be optimized, while other modules, including the celeb basis, face

encoder $F$, CLIP transformer $\tau_{\text{text}}$, and UNet $\epsilon_\theta$ are fixed. Thus, the original composition abilities of the trained text-to-image network are well-preserved, avoiding the forgetting problem. Since we only have a single photo, we use color jitter, random resize, and random shift as data augmentations on the supervision to avoid overfitting. Notice that, we find our augmentation method can work well even though there are no regularization datasets which is important in previous methods [10, 7, 8], showing the strong learning abilities of the proposed methods. Since the proposed method only involves a few parameters, it only takes almost *3 minutes* for each individual on an NVIDIA A100 GPU, which is also much faster than previous methods.

**Testing.** After training, only two groups of coefficients $A_1$ and $A_2$ applied to the principal celeb basis components need to be saved. In practice, the number of principal components of each group is $p = 512$, coming to only *1024 parameters* and *2-3KB* storage consumption for half-precision floatings. Then, users can build the prompt with multiple action description prompts (*e.g.* "*A photo of $V^*$ is playing guitar*") to synthesize the satisfied images as described in Fig. 4.

**Multiple Identities Joint Optimization.** Most previous methods only work on a single new concept [9, 10], Custom Diffusion [8] claim their method can generate the images of multiple new concepts (*e.g.* the sofa and cat). However, for similar concepts, *e.g.* the different person, their method might not work well as in our experiments. Besides, their method is still struggling to learn multiple ($> 3$) concepts altogether as in their limitation. Differently, we can learn multiple identities (*e.g.* 10) at once using a shared MLP mapping layer as in Fig. 4. In detail, we simply extend our training images to 10 and jointly train these images using a similar process as single identities. Without a specific design, the proposed method can successfully generate the weight of each identity. After training, our method can perform interactions between each new identity while the previous methods fail to as shown in the experiments. More implementation details are in the appendix.

## 4 Experiments

### 4.1 Datasets and Metrics

**Datasets.** We conduct experiments on the self-collected $2K$ synthetic facial images generated by StyleGAN [2]. Using synthetic faces as input for assessing the effectiveness of generative models, one can ease the reliance on the dataset foundation of the initial pre-trained text-to-image model. We also perform some experiments on the real photo of the individual, more results and comparisons are shown in the supplemental materials.

**Metrics.** First, we assess the performance of the generated images by utilizing objective metrics. For instance, we calculate the consistency between the prompt and generated image through CLIP score [4], which is denoted as "**Prompt**" in tables. Additionally, ensuring identity consistency and clarity of facial features are crucial aspects of our task. Therefore, we evaluate identity similarity using a pretrained face recognition encoder [14] and mark it as "**Identity**". Furthermore, to demonstrate the rationality behind generation, we also calculate the rate of successful face detection ("**Detect**") via a pretrained face detector [14]. Lastly, user studies are conducted to evaluate text-image alignment along with identity and photo qualities. The images having the highest text-alignment scores from a single inference batch are used compare the peak qualitative performance of all the methods.

### 4.2 Comparing with State-of-the-Art Methods

We compare the proposed method with several well-known state-of-the-art personalization methods for Stable Diffusion [1], including DreamBooth [10], Textual-Inversion [7] and Custom Diffusion [8]. As shown in Fig. 5, given one single image as input, we evaluate the performance of several different types of generation, including the simple stylization, concept combination abilities, and two new concept interactions. Textual inversion tends to overfit the input image so most of the concepts are

Table 1: Quantitative comparisons.

| Methods | Objective Metrics↑ | | | User Study↑ | | | #Params↓ | Time↓ (min) |
|---|---|---|---|---|---|---|---|---|
| | Prompt | Identity | Detect | Quality | Text | Identity | | |
| Textual Inversion [7] | 0.1635 | **0.2958** | **92.86%** | 2.23 | 1.88 | 2.55 | 1,536 | 24 |
| Dreambooth [10] | 0.2002 | 0.0512 | 54.76% | 3.32 | 3.70 | 2.75 | $9.83 \times 10^8$ | 16 |
| Custom Diffusion [8] | **0.2608** | 0.1385 | 80.39% | 3.31 | 3.55 | 2.96 | $5.71 \times 10^7$ | 12 |
| Ours | 0.2545 | 0.2072 | 84.78% | **3.47** | **4.01** | **3.37** | **1,024** | **3** |

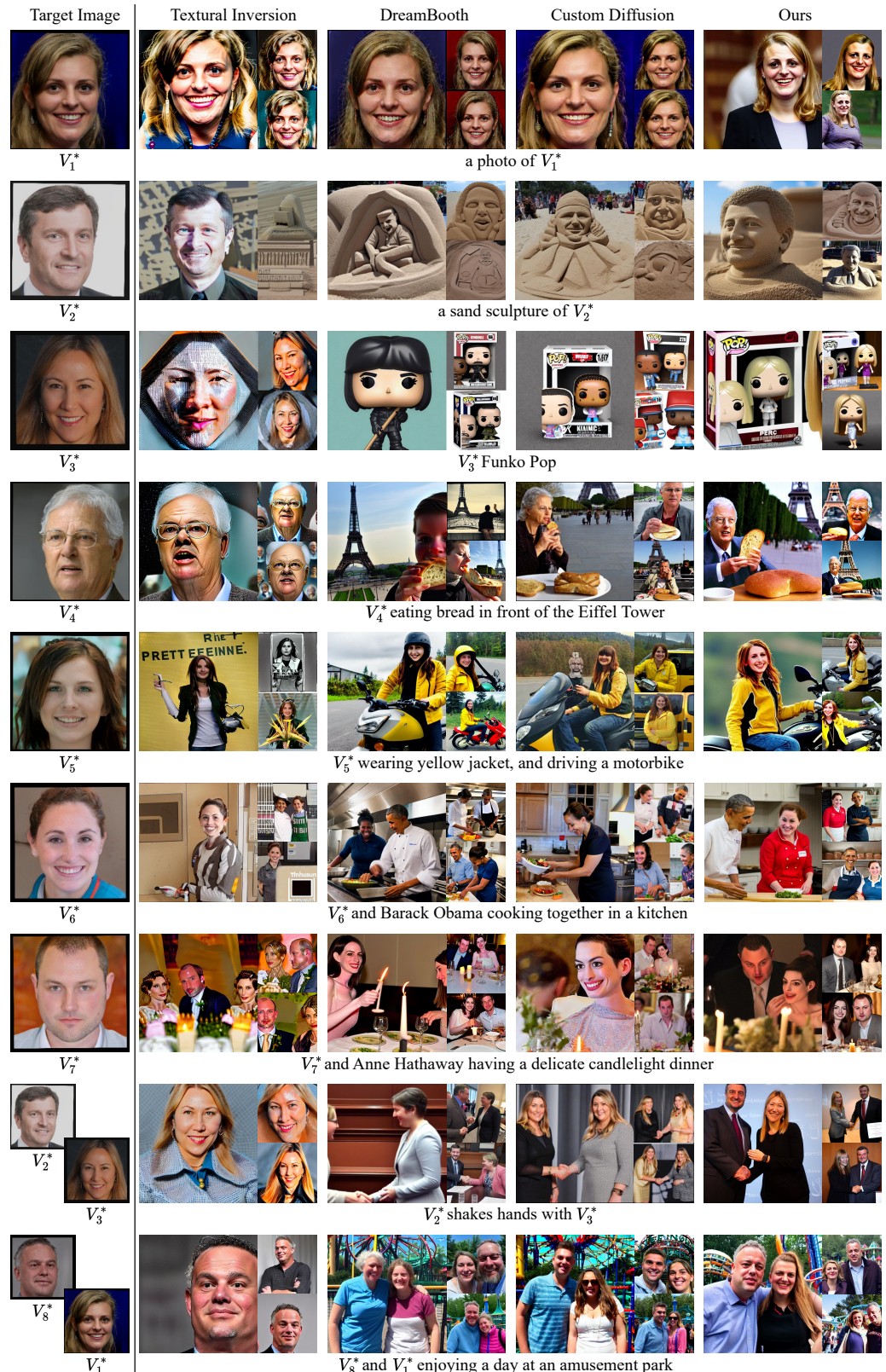

Figure 5: We compare several different abilities between our method and baselines (Textural Inversion [7], Dreambooth [10], and Custom Diffusion [8]).

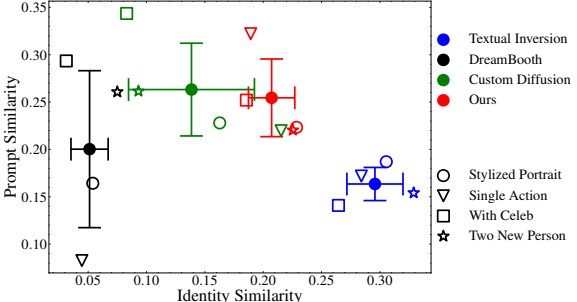

Figure 6: Numerical analysis in terms of the prompt and identity similarity on four prompt types.

| Methods | Prompt↑ | Identity↑ | Detect↑ |
|---|---|---|---|
| w/o celeb basis | 0.1386 | **0.2528** | 69.28% |
| w/ 350 names | 0.2214 | 0.2023 | 69.28% |
| w/o filter | 0.1939 | 0.2037 | 80.62% |
| w flatten | 0.2026 | 0.1873 | 80.39% |
| $p = 64$ | 0.2247 | 0.1061 | 76.47% |
| $p = 256$ | 0.1812 | 0.0656 | 60.13% |
| $p = 768$ | 0.1380 | 0.0836 | 47.06% |
| w/o $F$ | 0.1914 | 0.1896 | 55.56% |
| w/o aug. | 0.2083 | 0.1931 | 75.16% |
| Ours (single) | 0.2234 | 0.2186 | 81.05% |
| Ours (joint) | **0.2545** | 0.2072 | **84.78%** |

Table 2: Ablation studies.

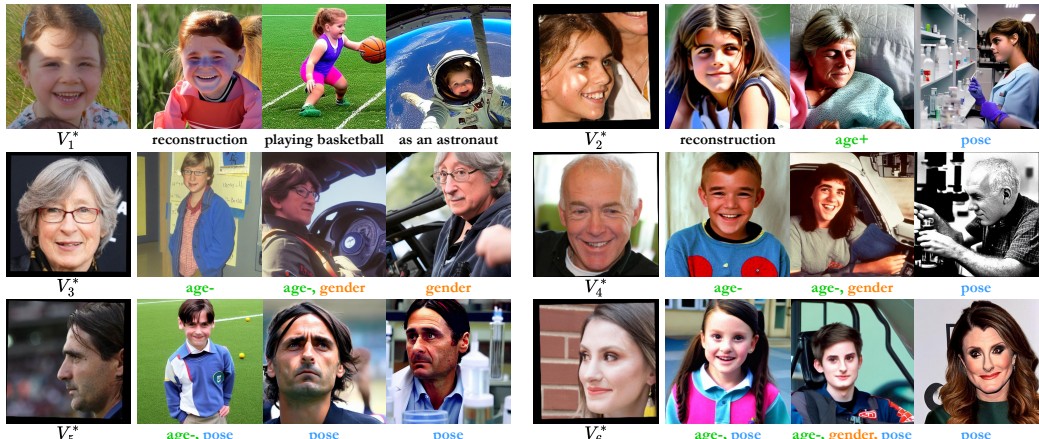

Figure 7: Our method is not only robust to large age, gender, head pose, and light variants in the inputs, but also has good ability of controlling age, gender, and head pose in the outputs.

forgotten. Although dreambooth [10] and custom diffusion [8] can successfully generate the human and the concept, the generated identities are not the same as the target image.

Besides visual quality, we also perform the numerical comparison between the proposed method and baselines in Tab. 1. From the table, regardless of the over-fitted Textual-Inversion (highest identity and detect scores but lowest prompt score), our method shows a much better performance in terms of identity similarity and the face detection rate and achieves similar text-prompt alignment as Custom Diffsuion [8]. We also plot the generated results on four detailed types in Fig. 6, where the proposed method shows the best trade-off. Notice that, the proposed method only contains very few learning-able parameters and optimizes faster.

Moreover, since identity similarity is very subjective, we generate 200 images from different identities to form a user study. In detail, we invite 100 users to rank the generated images from one (worst) to five (best) in terms of visual quality, prompt alignment, and identities, getting 60k opinions in total. The results are also shown in Tab. 1, where the users favor our proposed method.

### 4.3 Results on Age, Gender, Head Pose, and Expression

We qualitatively evaluate the robustness of our method to input age, gender, head pose, and light variants in Fig. 7. Besides, modifying the text prompts can edit the age, gender, and pose of the results. We further provide the expression controlling results in the appendix, along with the comparisons with Face Composer [51]. Thanks to the identity disentangling ability of the face recognition [14] network, the learned representations are highly relevant to the input identities while irrelevant and insensitive to other attributes like age, pose, expression, etc. Furthermore, the learned representation as an interpolation of celeb basis inherits the good characteristic of those "recognized" celeb names, which corresponds to large amount of image-text paired samples [3] used for pretraining Stable Diffusion [1].

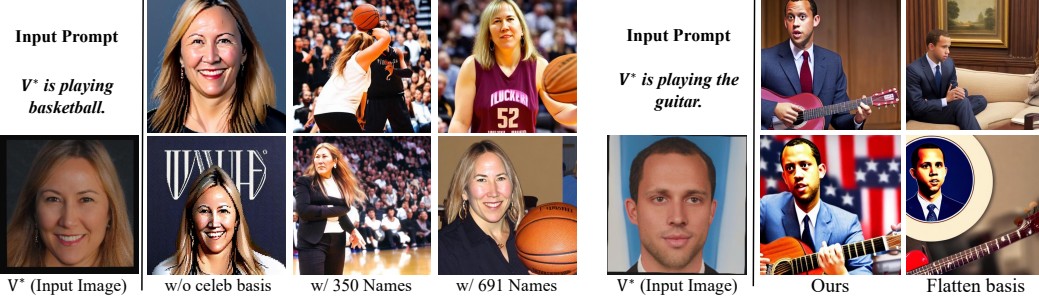

(a) # of names in celeb basis.    (b) First and last name basis *v.s.* flatten basis.

Figure 8: Ablation studies on building celeb basis.

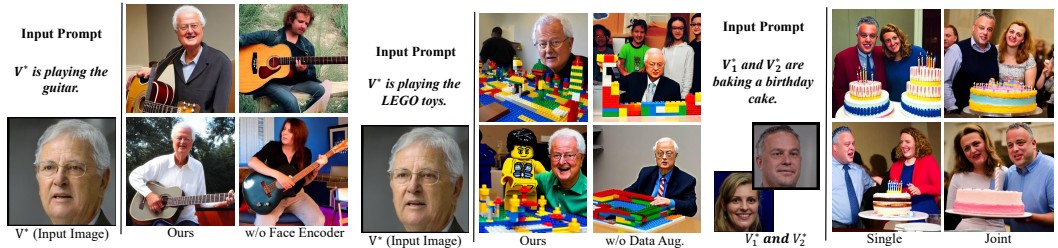

(a) Face encoder.   (b) Data augmentation   (c) Single *v.s.* joint optimization

Figure 9: Ablation studies on coefficient optimization.

## 4.4 Ablation Studies

To further evaluate the sub-modules of our proposed method in both building celeb basis (Sec. 4.4.1) and the proposed new personalization method (Sec. 4.4.2), we start from the default settings ( 'Ours (single)' in Tab. 2) of our method, conducting the ablation study by separately removing each submodule or using a different setting as follows. Due to the space limitation, we give more visual results in the appendix.

### 4.4.1 Ablation Studies on Celeb Basis

**# of names in celeb basis.** We evaluate the influence of the names to build a celeb basis. In extreme cases, if there is no name and we directly learn the embedding from the face encoder $F$ (w/o celeb basis), the model is overfitted to the input image and can not perform the concept combination. With fewer celeb names (w/ 350 names), the generated quality is not good as ours baseline (single) as in Fig. 8a. Besides, the quality of the celeb basis is also important, if we do not filter the names (w/o filter), the performance will also decrease as in Tab. 2.

**Flatten basis *v.s.* first and last name basis.** In the main method, we introduce our celeb basis as the *first and last name basis* since each name embedding does not have the same length. We thus involve a more naive way by flattening all the embeddings to build the basis (w/ flatten). As shown in Fig. 8b and Tab. 2, the generated images of our first and last name basis understand prompts better.

**Choice of reduction dimension $p$.** We also evaluate the influence of the number of coefficients $p$. Considering the 768 dimensions of the CLIP text embedding, we vary $p$ ranging in $\{64, 256, 512, 768\}$. Note that $p = 768$ can be also considered as 'w/o PCA' since the dimension of text embedding is 768. As shown in Tab. 2, the best result is obtained from the baseline choice ($p = 512$) and we show the differences in the generated images in the appendix.

### 4.4.2 Ablation Studies on Coefficients Optimization

**W/o face recognition encoder $F$.** Naively, we can optimize the coefficients $A_1, A_2$ of the celeb embeddings from back-propagation directly. However, we find the search space is still large to get satisfied results as shown in Fig. 9a and Tab. 2 (w/o $F$). So we seek help from the pretrained face encoder, which has more discriminative features on the face.

**W/o data augmentation.** Since there is only one single image as the tuning-able sample, we perform some data augmentations as introduced in Sec. 3.2. However, if we remove these augmentations, the generated face becomes vague and unnatural as shown in Fig. 9b, and the identity of the generated samples is also decreased as shown in Tab. 2.

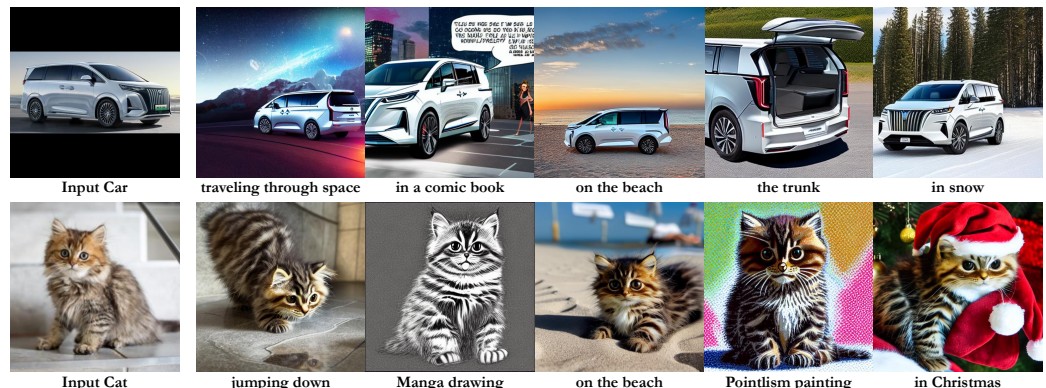

| Input Car | traveling through space | in a comic book | on the beach | the trunk | in snow |

| Input Cat | jumping down | Manga drawing | on the beach | Pointlism painting | in Christmas |

Figure 10: Through a simple replacement on the encoder and text basis, our method can be extended to more general classes, *e.g.* car and cat.

**Single *v.s.* joint training.** Our method supports joint training of multiple identities using a single MLP, we evaluate the differences between single training/joint testing and joint training/joint testing. As shown in Tab. 2 and Fig. 9c, although training individually can also perform some interactions between the two-person, training the images jointly improve the robustness and reduces the overfitting risk compared with single training, resulting in slightly better results.

## 5   Conclusion & Discussion

We propose a new method to personalize the pre-trained text-to-image model on a specific kind of concept, *i.e.* the human being, with simply single-shot tuning. Our approach is enabled by defining a basis in the domain of the known celebrity names' embeddings. Then, we can map the facial features from the pre-trained face recognition encoder to reconstruct the coefficients of the new identity. Compared with the previous concept injection method, our method shows stronger concept combination abilities, *e.g.* better identity preservation, can be trained on various identities at once, and can produce results where the newly-added humans interact with each other. Besides, the proposed method only requires 1024 parameters for each person and can be optimized in under 3 minutes, which is also much more efficient than previous methods.

**Limitations.** Although our method can successfully generate the images of new identities, it still occurs some limitations. First, the human faces generated by original stable diffusion [1] intrinsically contain some artifacts, causing somewhat unnatural performance of the proposed method (see our appendix). It might be solved by a more powerful pre-trained text-to-image model (*e.g.* Imagen [5], IF [52]) since they can generate better facial details. Second, we only focus on human beings currently. It is also interesting to build the basis of other species, *e.g.* cars, and cats. To this end, we provide some rough experiments (whose implementation details are introduced in the appendix) of extending our method to cars and cats, as shown in Fig. 10. The results show that our method can also generate text-aligned images with good quality without bells and whistles. But the simple try could be improved through using a stronger encoder or crawling more text to construct basis. Third, although the text alignment scores evaluate the quantitative performance, for the editing ability of age, expression, and pose, we do not have a metric result yet. As supplement, we will use facial feature extractors to quantitatively compare the human identity personalized tasks in the future. Fourth, similar to the inference-only method FastComposer [51], we have also tried to train a model on 50k identities (20 faces per identity without text labels) as an encoder-based method. However, it is hard to converge in a short GPU time within a batch size of eight. We think it might be because we need much more data and computing resources to train a unified or general facial model and we will explore it in the future.

**Ethics Consideration and Broader Impacts.** We propose a new method for model personalization on the human face, which might be used as a tool for deepfake generation since we can generate not only the human face but also interactions between people. However, this prevalent issue is not limited to this approach alone, as it also exists in other generative models and content manipulation techniques. Besides, our personalization person generation method can also ablate the abilities of the erasing concept methods [53] and other deep fake detection methods [54].

# 6 Acknowledgements

This work was supported in part by the National Natural Science Foundation of China under Grant 61976231 and in part by the Guangdong Basic and Applied Basic Research Foundation under Grant 2023A1515012853.

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

# A  More Qualitative Results

## A.1  Evaluation on Faces in a Wider Range

Besides evaluating the results on StyleGAN synthetic faces as in the main paper, we compare our method with state-of-the-art methods on a wide range of human face images, including the real word faces, the faces from different races, and the interactions of the same gender. These face images are from VGGFace2 [55] and collected from the Web.

**Single Person in an Image.** Fig. 11 shows the evaluation results on a single person's personalization for real identities. Concretely, Row 1-2 shows the ability to generate diverse images of different methods. The baselines Textual Inversion [7], DreamBooth [10], and Custom Diffusion [8] can generate identity-consistent results with the single prompt 'A photo of $V^*$', but the light condition and skin texture seem not natural. DreamBooth and Custom Diffusion fail to disentangle the background of the target inputs. The stylization results are shown in Row 3-4, where Textual Inversion preserves the target identities but fails to change the style, DreamBooth, and Custom Diffusion generate the images of the corresponding style but lose the identity information, and our method preserves the identity and completes the stylization as given prompts.

**Multiple Persons in an Image.** We give the comparison of generating multiple-person interaction in Fig. 12. Similar to the single-person results, the proposed method shows a good concept combination ability among different persons (including the newly added and the original celebrities) and can serve as a novel concept to communicate with other humans.

## A.2  Additional Results Comparing with Concurrently Work: FastComposer

Different from the methods [7, 10, 8] that the finetuning time cost for each identity is below 30 minutes, FastComposer [51] needs to be pre-trained on a large human dataset, requiring 150k steps with batch size 128 on 8 NVIDIA A6000 (48GB) GPUs. So as its performance is restricted by the dataset as in their limitation. Differently, our method requires only 400 steps with batch size 2 on a single NVIDIA A100 (40GB) GPU. After the pre-training stage, FastComposer can generate images based on the given single image and prompt like other personalized methods.

To further demonstrate the efficacy of our method, we evaluate the qualitative performance of our method compared with FastComposer. Our method outperforms FastComposer on single action controlling, the interaction between two persons, and expression controlling. The detailed comparisons are as follows.

**Action of Single Person.** We first evaluate the performance of single persons' personalization. As illustrated in Fig. 13 and Fig. 14, we show the synthesized results under six scenarios in each figure.

In Fig. 13, given the prompt 'A photo of $V_1^*$', FastComposer generates an unnatural result. In the remaining scenarios, the light seems very disharmonious in the images synthesized by FastComposer. Besides, in these results, only the upper chest and head can be generated, where the human body and limbs fail to appear, revealing an over-fitting issue of FastComposer.

In Fig. 14, FastComposer meets the same issue. Although the key objects and face identity are almost consistent with the input prompts, e.g. LEGO toys, fighter jet, and yellow jacket, the results from FastComposer seem to be a rude combination of the objects and face. Besides, their methods have a huge issue with concept forget. Some important prompts like 'sweater' and 'motorbike' are ignored by FastComposer.

We also highlight the faces generated by FastComposer share very similar expressions from the original input image, i.e. dilated pupils, wrinkles on the forehead, and mouth shape, which means this method is overfitted on the input image. Differently, our method generates identity- and prompt-consistent results, appearing natural and realistic.

**Interaction between Two Persons.** Fig. 15 illustrates the interaction and common action of two persons in a single image. Sharing the similar aforementioned over-fitting problem, FastComposer only generates the face part of the human. As for the actions including 'shaking', 'playing', 'sitting', and 'baking', FastComposer fails to generate the correct behaviors, where the faces usually occupy the most part of the images. The problem of invariant expression also occurs in FastComposer.

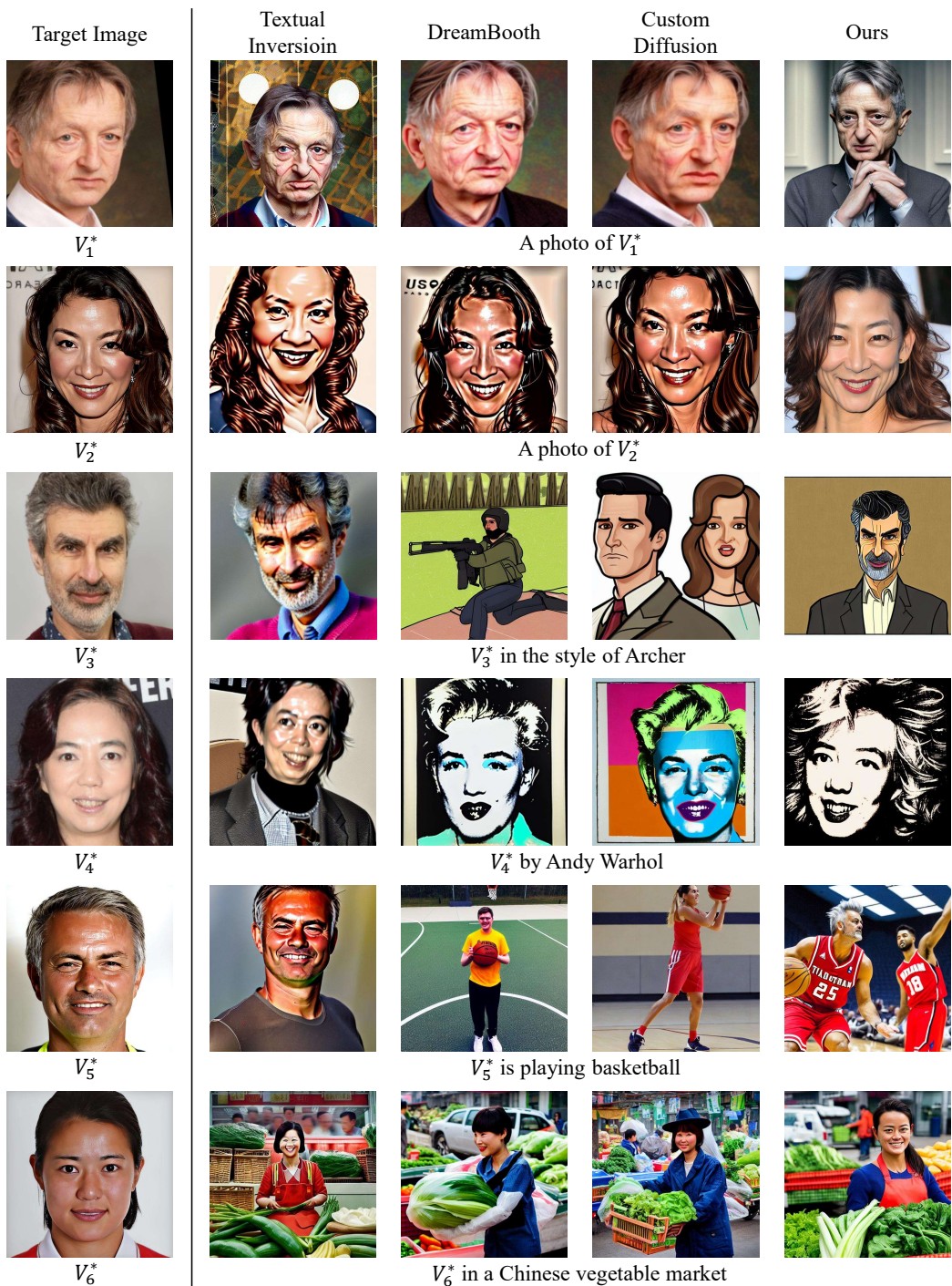

Figure 11: Single person's personalization result for **real** identities.

**Expression Controlling.** Considering that the above FastComposer results struggle to generate the different expressions, we conduct a face expression controlling experiment to validate the abilities of our method and FastComposer. We use the following prompts to change the expression of the target input image:

- $V^*$ is smiling with happiness
- $V^*$ is crying with sadness

- $V^*$ has an anger expression
- $V^*$ looks very surprised, looking at a gift box
- $V^*$ is shaking with fear scared by a candle lighting, scared expression
- $V^*$ has a tired expression, headache, uncomfortable, a frown creased her forehead

As shown in Fig. 16, the proposed method can successfully generate different expressions under the text prompts, which shows the advantage of the proposed method.

### A.3 Uncurated Results

We show the uncurated results of our method in Fig. 17, which demonstrate that our method can generate stable and text-aligned images without being cherry-picked.

## B Implementation Details

### B.1 Celeb Images Generated by Stable Diffusion

In general, the celeb images generated by Stable Diffusion [1] and those realistic ones collected from Google Images seem very similar, but with a little difference on facial details. As seen in Fig. 18, we compare the real celeb images collected from Google Images with the fake ones synthesized by Stable Diffusion. Overall, the real and fake images closely resemble the same individual. However, certain facial features in the fake images may be exaggerated, which can constrain our method's performance and explain why the celebrity profiles in the outcomes generated by both baseline models and our approach do not precisely match their real counterparts.

### B.2 Collecting and Filtering Celeb Names

After crawling about 1,500 celebrity names, to filter out the names that have the ability to generate prompt-consistent identities and interacting with other celebs, we feed three types of prompts, i.e. 'A photo of $V_c$', '$V_c$ is playing the guitar', and '$V_c$ talks with Barack Obama', to the Stable Diffusion [1] for synthesizing image results, where $V_c$ indicates the celeb name (e.g. 'Anne Hathaway'). Fig. 19 shows the examples that satisfy the filtering condition and the ones that fail to generate reasonable results. Consequently, 691 names pass the checking, which can be tokenized and encoded into $m = 691$ celeb embedding groups.

The statistics of these 691 names are shown in Fig. 20. Specifically, we count the frequency of each celeb name in the LAION-2B dataset text samples. LAION-2B is a subset of LAION-5B [3] where Stable Diffusion [1] is pretrained. After sorting the celeb names according their frequency, we obtain Fig. 20, where the x-axis is the index of the name, the y-axis is the appearing time of this name in the dataset. This plot indicates that our collected celeb names correspond to thousands image-text paired samples in average. There are only 25 names whose frequencies are less than 100 and 173 names whose frequencies are larger than 10000. This frequency distribution shows that the distribution of the celeb names is relatively fair.

### B.3 Building Celeb Basis based on PCA

We only keep the first and second embeddings of each celeb embedding group, resulting in the first name embedding set $C_1$ and the second name embedding set $C_2$, where $C_1, C_2 \in \mathbb{R}^{m \times d}$. Then, for simplicity, we omit the subscript of $C_k$, using $C$ to indicate any one of $C_1$ and $C_2$.

**Calculation of Mean.** Considering $C$ may have repetitive embeddings (each row corresponds an embedding vector), for each $C$, we first remove the duplicate rows that come from the same token to make sure each token only occurs once at most. Then we calculate the mean $\overline{C} \in \mathbb{R}^d$ across the second dimension of $C$ as mentioned in the main paper.

**PCA.** The PCA algorithm has many different coding implementations in practice. In our method, we use Singular Value Decomposition (SVD) to skip the calculation of covariance matrix in PCA. Please refer to [13, 56] for more detailed theoretical demonstration and coding techniques. Due to the built celeb basis is not optimized, the PCA process only needs to be computed once during the training stage.

**B.4   Training Recipe**

We train the MLP with a learning rate of 0.005 and batch size of 2 on a single NVIDIA A100 GPU. The training augmentation includes horizontal flip, color jitter, and random scaling ranging in $0.1 \sim 1.0$. For single identity training, the optimization costs 400 steps, taking $\sim 3$ minutes. For 10 identities joint training, we found that training 2,500 steps is enough, taking $\sim 18$ minutes (averaged about 250 steps and 108 seconds for each identity). The text prompts for training include:

- A photo of a face of $V^*$ person
- A rendering of a face of $V^*$ person
- The photo of a face of $V^*$ person
- A rendition of a face of $V^*$ person
- A illustration of a face of $V^*$ person
- A depiction of a face of $V^*$ person

## C   More Ablation Studies

**W/o filtering celebrity names.** As mentioned in Section B.2, Stable Diffusion fails to generate correct images from some celeb names. In our method, we manually drop these bad samples. If this filtering process canceled, the interaction ability of the learned identities drops a lot, as shown in Fig. 21a.

**Variants of reduction dimension** $p$**.** In our method, the PCA reduction dimension $p$ $(p < k)$ controls the degrees of freedom for modulating the variance applied to the celeb basis mean. Considering that $k = 768$ in the CLIP [4] text encoder, we conduct a series experiments to study the choice of $p$. Four values of $p$ are chosen, including $64, 256, 512, 768$. Note that $p = 768$ means PCA is not used. Larger $p$ costs more memory storage. As shown in Fig. 21b, chosing $p = 512$ comes to the best identity similarity, which is consistent with the quantitative results in our main paper.

**Variants training images per identity** $N$**.** Our method focuses on one-shot identity personalization problem (i.e. training from a single photo to represent a newconcept). To verify the influence of training images count per identity, we provide the results in Fig. 22, where the inputs varies in view and lighting condition. It shows that when we use more images, the face quality can also be slightly improved, and the synthesized results are already pretty good when $N = 1$.

## D   Extending to More General Classes

Our method have potentials to be extended to non-face classes. Here, we provide some rough experimental results in Fig. 10. For non-face samples, we trivially replace the face recognition encoder with a pre-trained ResNet50 (trained on ImageNet-1K) and construct the "car" (or "cat") basis by collecting typical types of cars (or breeds of cats) in phrases, *e.g.* 'Volkswagen Passat' and 'Mercedes-Benz A-Class' (or 'Scottish Fold' and 'Abyssinian' for cats). Without other bells and whistles, our method generates text-aligned images with good quality of cars and cats. In the future, we would try a better general encoder (to replace ResNet50) and crawl a larger basis for non-face classes, extending our method for more general subjects using our well-collected basis names.

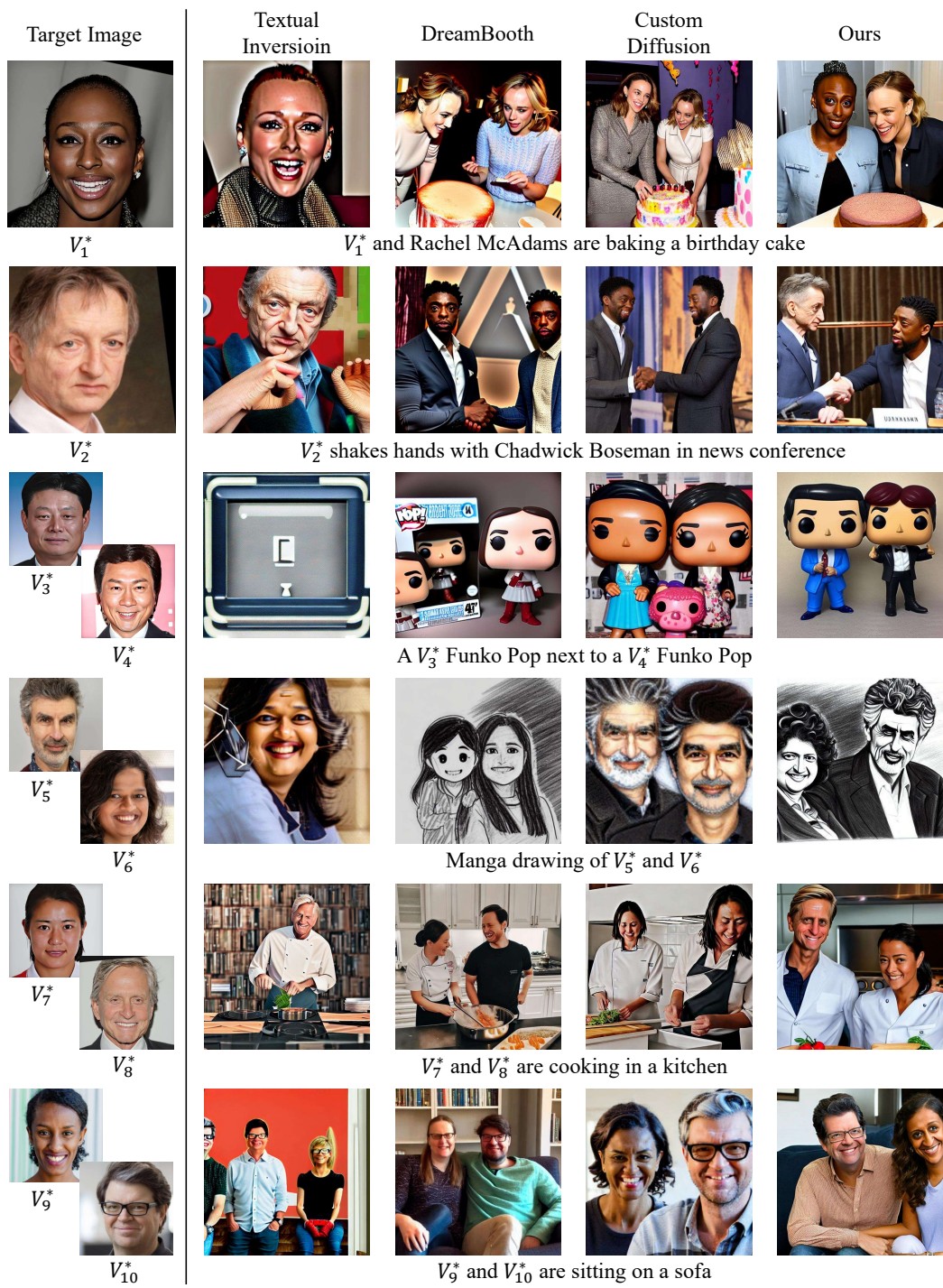

Figure 12: Multiple persons' personalization result for **real** identities.

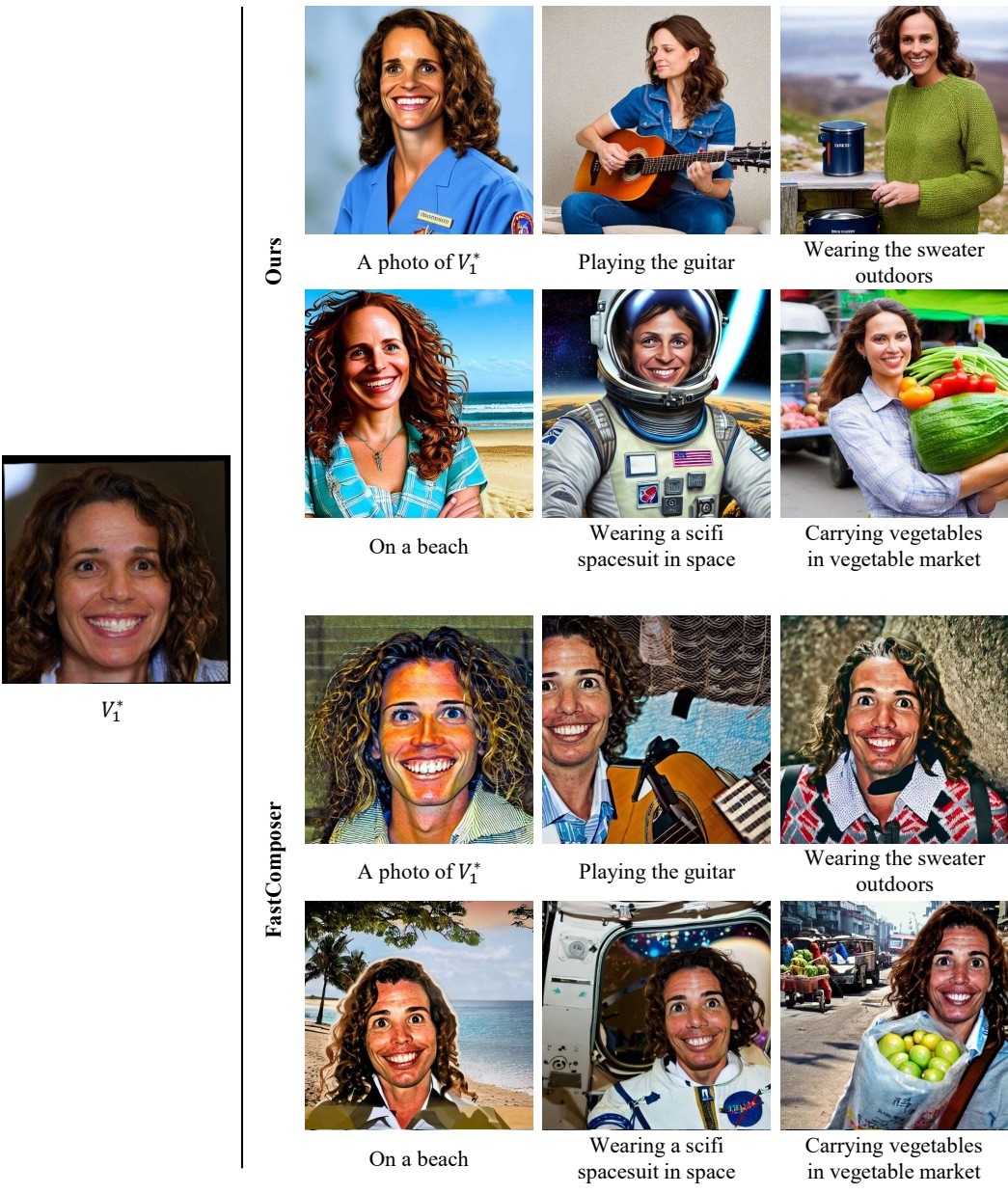

Figure 13: Additional single person's personalization results comparing with FastComposer [51].

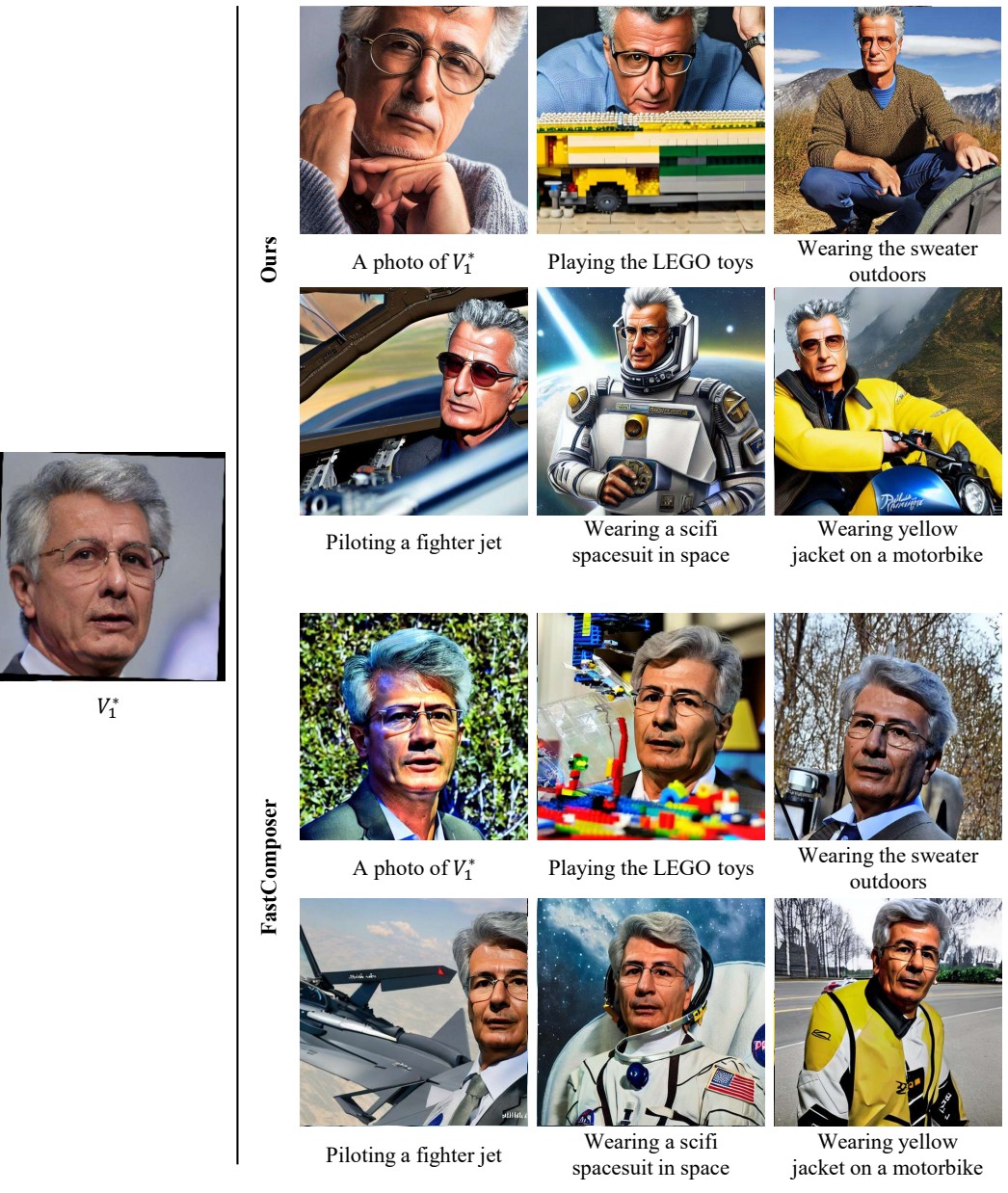

Figure 14: Additional single person's personalization results comparing with FastComposer [51].

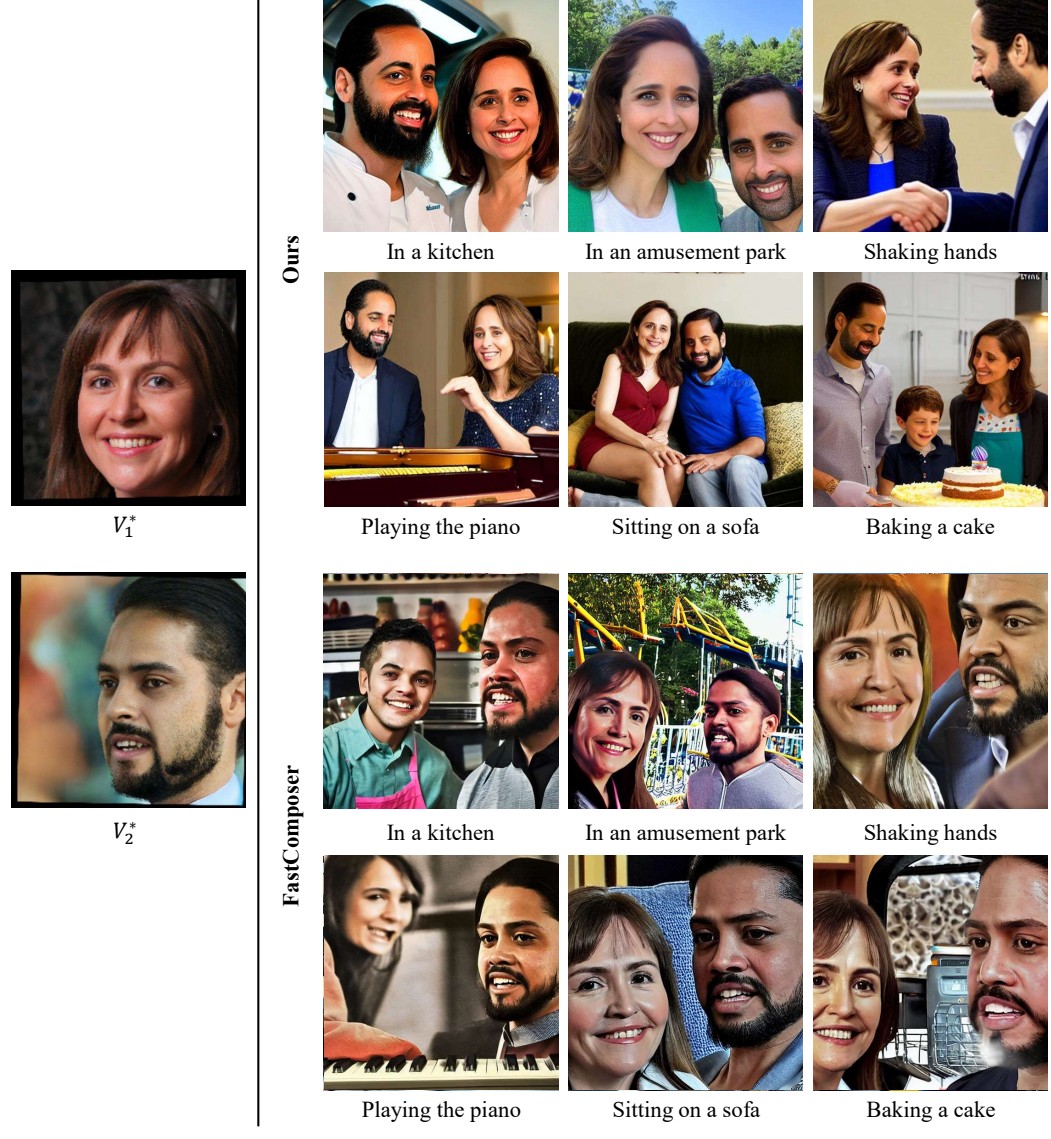

Figure 15: Additional multiple persons' interaction personalization results comparing with FastComposer [51].

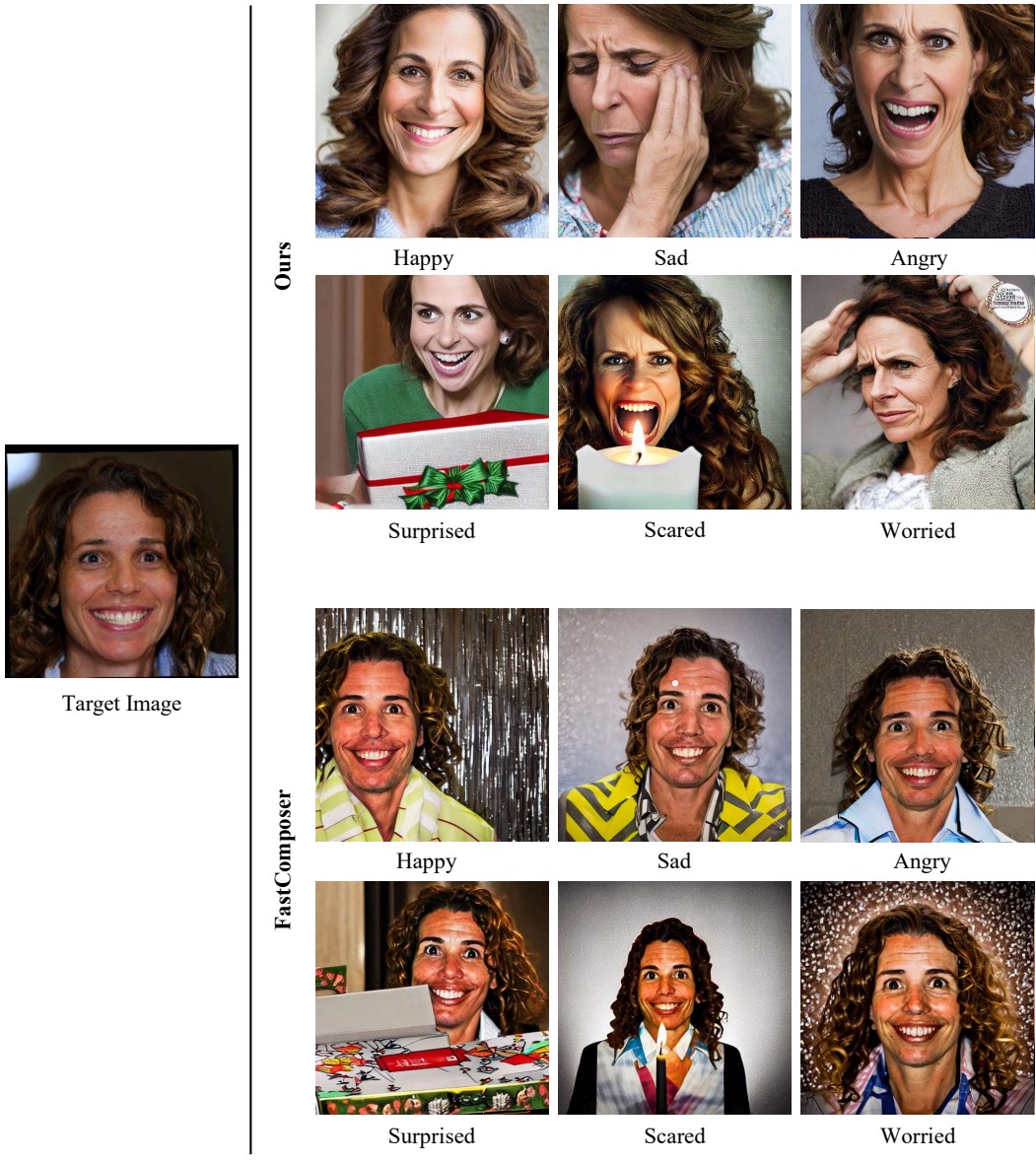

Figure 16: Comparing with FastComposer [51], our method has a better ability of controlling face expression.

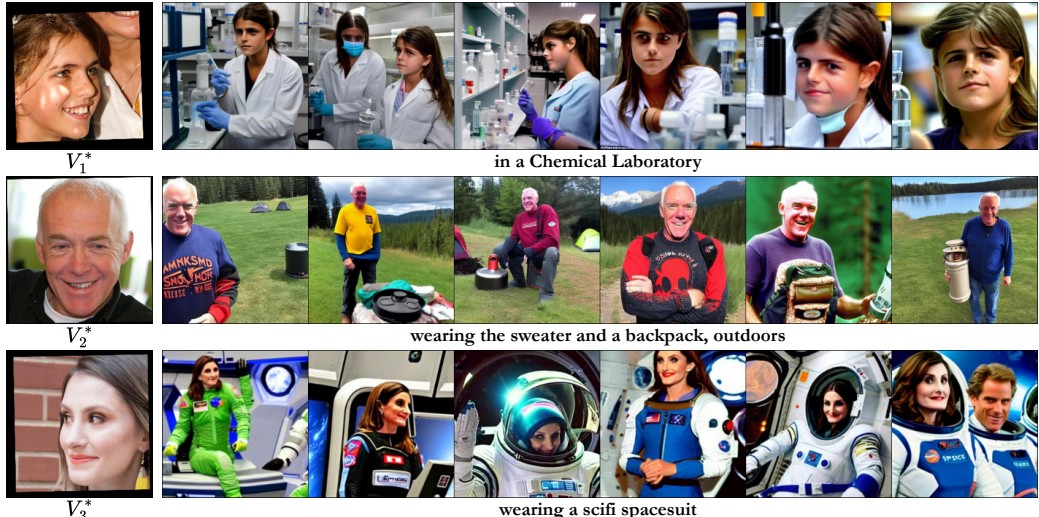

$V_1^*$

in a Chemical Laboratory

$V_2^*$

wearing the sweater and a backpack, outdoors

$V_3^*$

wearing a scifi spacesuit

Figure 17: Uncurated results in a testing batch.

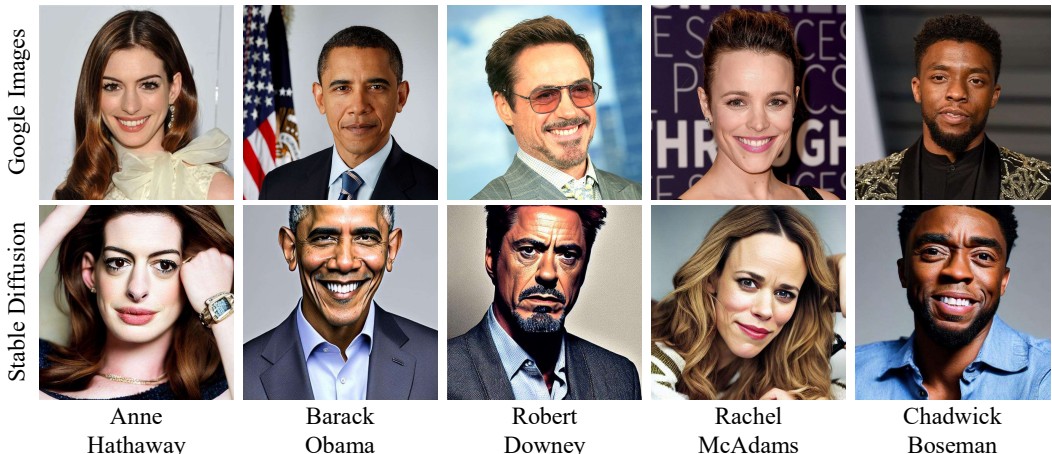

Google Images

Stable Diffusion

Anne Hathaway    Barack Obama    Robert Downey    Rachel McAdams    Chadwick Boseman

Figure 18: Stable Diffusion has the ability to generate identity-consistent celeb images. But in the aspect of facial details, the celeb images synthesized by Stable Diffusion may exaggerate some facial features compared with the real ones collected from Google Images.

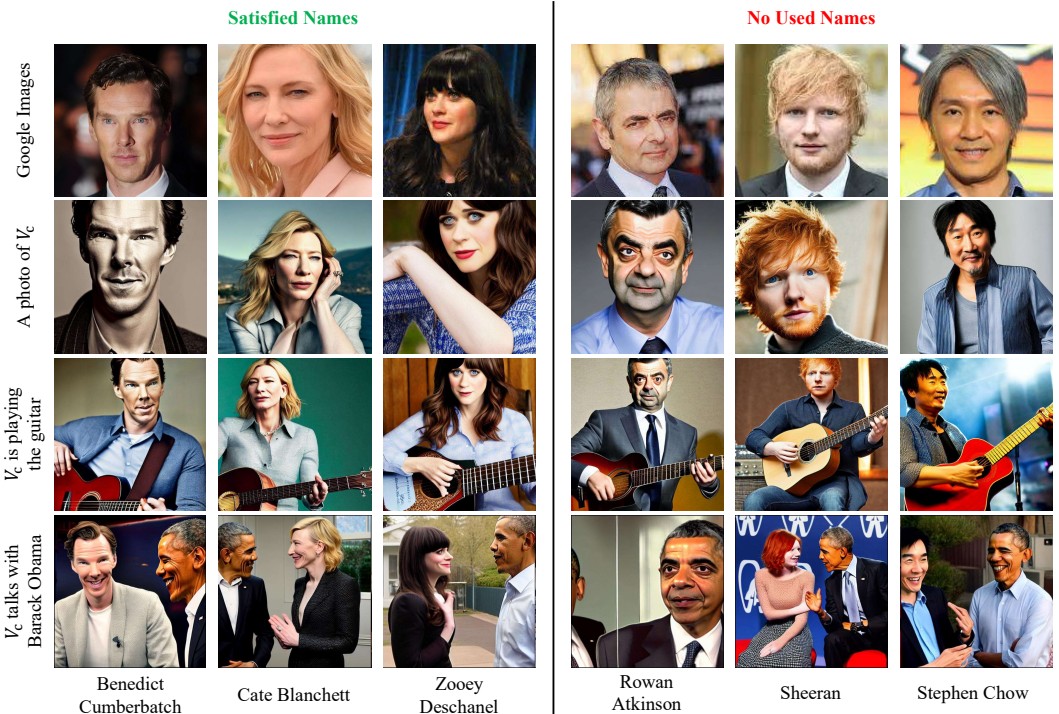

Figure 19: We filter out the celeb names that can generate prompt- and identity-consistent images through Stable Diffusion [1]. Left: the satisfied names that can generate images share the same identity with Google Images results. Furthermore, these names have the ability to interact with objects or other satisfied celebs (e.g. 'Barack Obama'). Right: Stable Diffusion confuses the face of 'Rowan Atkinson' and 'Barack Obama' in row four. The gender of 'Sheeran' is mistaken as a girl in row four. The identity of 'Stephen Chow' has a large gap with that of Google Images results.

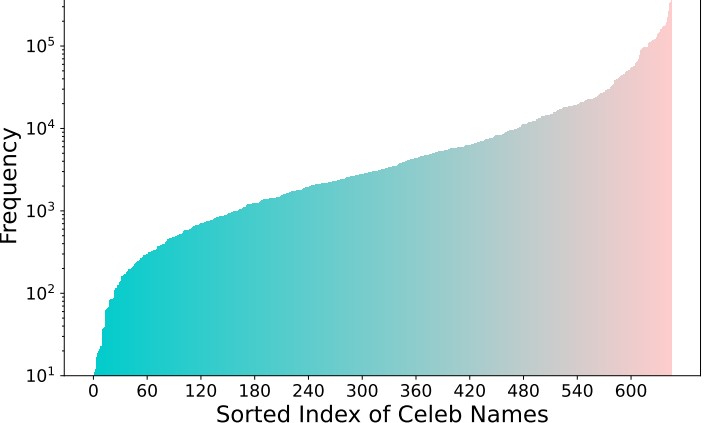

Figure 20: The frequency statistics of our used celeb names in LAION-2B [3] (a subset where Stable Diffusion trained).

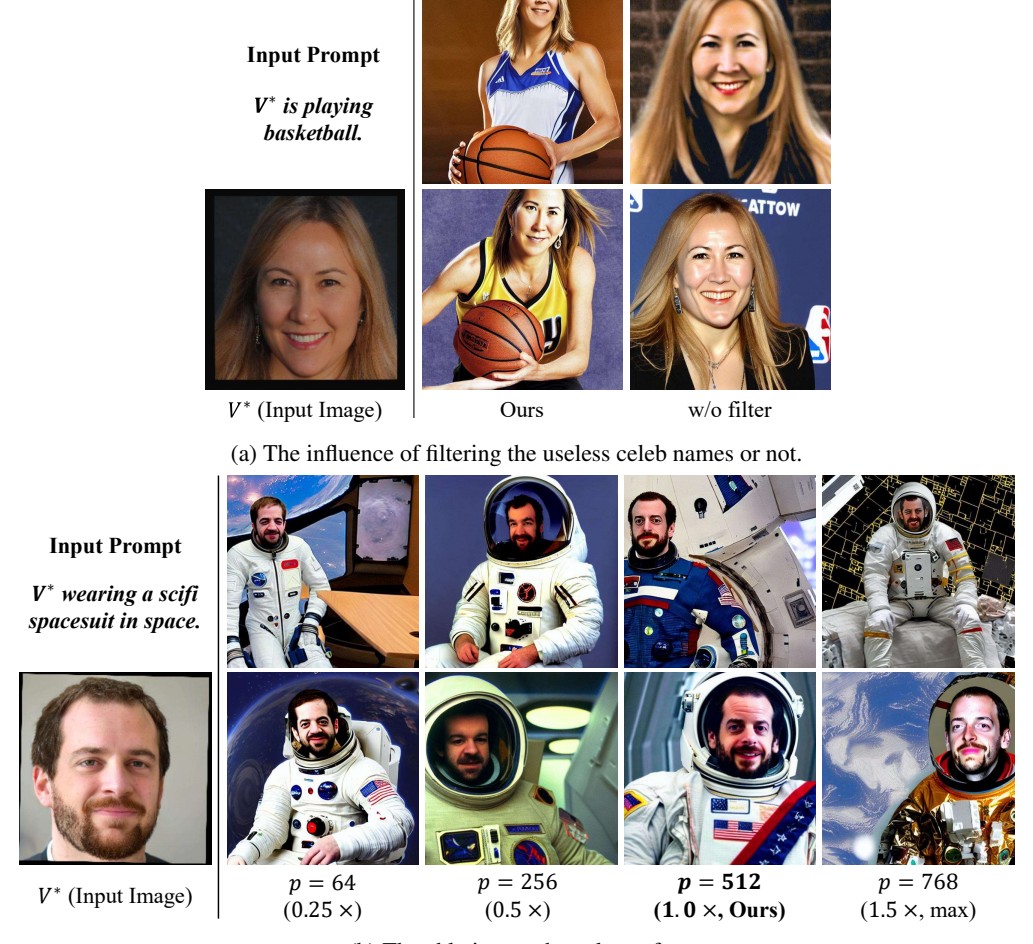

(a) The influence of filtering the useless celeb names or not.

(b) The ablation on the values of $p$.

Figure 21: Additional ablation studies on building celeb basis and the choice of reduction dimension $p$. Note that in our method, the text embeddings are 768-dimensions, which means $p = 768$ is equivalent to the method without PCA.

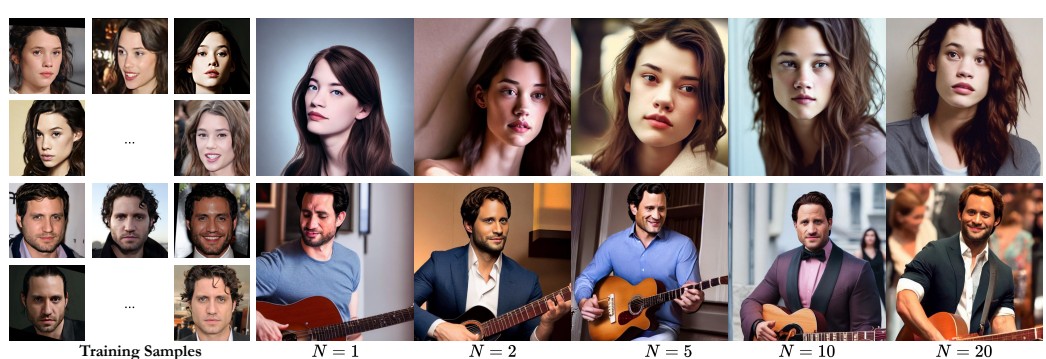

Figure 22: Additional ablation studies on the number of training images per identity $N$.

