# OpenReview forum: "Inserting Anybody in Diffusion Models via Celeb Basis"
_NeurIPS.cc/2023/Conference — NeurIPS 2023 poster_

### Official Review · Reviewer_RQjb · 2023-07-05

**Soundness:** 3 good
**Presentation:** 3 good
**Contribution:** 2 fair
**Rating:** 5
**Confidence:** 5

**Summary:**

This paper describes a new method for personalizing pre-trained text-to-image models to generate images of unique individuals using just one facial photograph. The method analyzes and builds a celeb basis from the embedding space of the pre-trained text encoder, generating the new identity's embedding by optimizing the weight of this basis and locking all other parameters. The proposed method shows better concept combination ability than previous methods and can learn several new identities at once. The authors plan to release their code.

**Strengths:**

1. The paper is well-written and easy to read.
2. It is an interesting approach to constructing a celeb basis from the text embedding space and optimzizing its weight to generate unique embeddings.
3. The results demonstrate the effectiveness of this approach and highlight the potential for generating highly customized images.

**Weaknesses:**

1. Using a Celeb name basis may limit the expressiveness of the personalized model, as the generated images may be constrained by a predefined set of celebrity names. However, it is unclear what distribution of face, age, and gender was used. This information is crucial to assessing the robustness and generalizability of the proposed method,

2. The paper mentions the importance of using PCA to find the Celeb basis. However, the paper seems to lack an ablation study without using PCA to evaluate the impact of this step on the performance of the personalized model. Without such a comparison, it is difficult to assess the role of PCA in the proposed method and whether it is necessary for achieving the desired results.

3. The generated faces in the proposed method, as shown in Figure 5 for the "Anne Hathaway" example, have low quality and distortion. It could be better to provide more analysis.


**Questions:**

see weakness

---

> ### Author Rebuttal · Authors · 2023-08-10
>
> # 5. Responses to R5 (RQjb)
>
> We sincerely appreciate for your careful reviews. Our responses to your questions are listed as follows:
>
> **R5Q1.** ***Using celeb name basis may constrain the representative ability? What is the distribution of used celeb names?***
>
> Our method is not constrained by the used celerity basis as shown in global response **GQ1**. The personalized results where the faces have large age, pose, expression, gender variants also demonstrate the strong representative ability of celeb basis, as stated in **GQ2**.
>
> The `Rebuttal PDF Figure 2` shows the distribution of used celeb names. Specifically, we count the frequency of each celeb name (total $691$) in the LAION-2B dataset text samples. LAION-2B is a subset of LAION-5B where Stable Diffusion [1] is pretrained. After sorting the celeb names according their frequency, we obtain `Rebuttal PDF Figure 2`, showing that the most of celeb names appear for $10^3\sim10^4$ times. Since collecting and validating the detailed gender, age, etc. attributes of our used celeb names is hard to be accomplished in short time, we leave more detailed distribution anlysis as the future work.
>
> **R5Q2.** ***No ablation study of PCA [13]?***
>
> Sorry for the unclear details. Actually, we have already ablated the PCA [13] in both `main paper Table 2` and `Supplement Figure 9`, where "$p=768$" indicates that there is NO PCA [13] for dimension reducation. Because the dimension of the extracted features from the CLIP text encoder equals to $768$. The existing annotation of $p=768$ in `main paper Table 2` may mislead the reviewer considering it still using a PCA [13] operation. We'll modify the related formulations and declare this more clearly in the paper.
>
> **R5Q3.** ***The synthesized celeb faces in `main paper Figure 5` seem distorted?***
>
> As clarified in `main paper Section 4.4 Limitation`, we have used Stable Diffusion [1] v1.4 and v1.5 to perform face synthesizing experiments, both of which are hard to generate prefect faces without extra large-scale finetuning. Furthermore, as seen in `Supplement Figure 7 and 8`, the Stable Diffusion [1] synthesized celeb faces may slightly distort from the realistic ones (collected from Google Images). These limitations of Stable Diffusion [1] may lead to the facial distortion of **celebrities** in the output results. That is, given a text prompt containing celeb name, the Stable Diffusion synthesized celebrity may have slight differences with the realistic one or the one in our "brains". This appearance does NOT originate from our method. Replacing Stable Diffusion [1] with a stronger diffusion model like Imagen [5], IF [51], SDXL can potentially alleviate this issue, which will be verifed as the future work.

---

> > ### Comment · Reviewer_RQjb · 2023-08-12
> >
> > Thanks for the rebuttal, which do help clarify some of the issues. Howver, I think Figure 2 in rebuttal pdf shows more examples only?

---

> > > ### Author Response · Authors · 2023-08-12
> > >
> > > Thanks for your carefully check. The `Figure 2 in Rebuttal PDF` (attached in the **Global Response** part) is a **histogram plot** showing the $691$ celeb names' frequency statistics in LAION-2B text samples (LAION-2B/5B dataset is used to train Stable Diffusion [1], which contains $\sim5.5$ billion image-text pairs), where the x-axis is the index of the name, the y-axis is the appearing time of this name in the dataset. This plot indicates that our collected celeb names correspond to thousands image-text paired samples in average. There are only $25$ names whose frequencies are less than $100$ and $173$ nemes whose frequencies are larger than $10000$. This frequency distribution shows that the distribution of the training is relatively fair.
> > >
> > > We would also be very willing to provide the detailed analysis of the distributions of each attribute, including age, gender, and race as you mentioned. However, we find it needs tones of time to crawl all the coorresponding celeb **___images___** from LAION dataset. For example, analyzing all the **text names** from LAION dataset will require $1$ day longer. And it will cost over $7$ days to download the corresponding **images** (just downloading, without analyzing the attributes), which seems impossible to be accomplished in such a short rebuttal time. We will also continue to download images and analysis these attributes and put them in the revised paper or our GitHub pages in the future to support our claim.
> > >
> > > Although without providing the age, gender, and race distribution, the qulitative results in `Rebuttal PDF Figure 1` along with the statistic in `Rebuttal PDF Figure 2` already demonstrate the robustness and generalizability against large age, gender, and pose variation of our method. And also as stated in **R5Q1**, we leave more detailed distribution anlysis as the future work to discover more about the bias in celeb basis.

---

### Official Review · Reviewer_Qr1K · 2023-07-05

**Soundness:** 3 good
**Presentation:** 3 good
**Contribution:** 3 good
**Rating:** 6
**Confidence:** 4

**Summary:**

The manuscript with title: Inserting Anybody in Diffusion Models via Celeb Basis presents an interesting and novel way to insert identity to pre-trained text-to-image models. The paper is well-written and has clearly delivered the main concept. This approach only requires a single image with few trainable parameters, which opens a new avenue for personalized text-to-image translation.

**Strengths:**

1. This paper presents an interesting and novel way to perform personalized text-to-image diffusion model with only one image. The key idea behind this is to map the image features to a pre-defined basis from Celebrity dataset, which is novel and works well.
2. The authors conducted a fairly amount of ablation studies, indicating the roles of each submodules, I appreciate that.
3. The results demonstrate improved visual quality compared to DreamBooth and Custom diffusion.


**Weaknesses:**

1. One key point that I'm concerned about is how sensitive is the method to different face positions or image quality. To construct the basis, I assume that the dataset faces are well-aligned and more uniform. However, in the real cases, those conditions not necessarily hold, a face can be tilted, not well aligned, however most of the examples in the paper and the supplementary material have the input face with good alignment. I wonder if the authors can share some face generation results with poor alignments. and how to handel this issue.

2. Also I noticed that quite a large amount of the inference results are also from the celebrity dataset (correct me if I got it wrong), would appreciate more results from more personalized images - also curious about the results that are different from the celebrity dataset, for example photos of babies.

3. Lack ablation studies on the number of images.

4. I assume this framework can be generalized to not face only text-to-image generation, as long as we have a good basis, can you elaborate on this? or showing some examples of non-face image generation, for example can this framework be generalized to personalized car generation?

**Questions:**

1. My key question is that if we want to apply this framework to a more generalized text-2-image generation, besides face generation, what do we need to do, how important is the celebrity basis. - lets say a simple experiments, what if we input a car image, where we can no longer accurately represented by the basis, what will the results be looking like?

**Limitations:**

1. The main limitation of this work is that it only focuses on face generation, with good alignment. This limitation may plaque the application of this work.

---

> ### Author Rebuttal · Authors · 2023-08-10
>
> # 4. Responses to R4 (Qr1K)
>
> Thanks for your helpful suggestions. And we respond to your questions as follows:
>
> **R4Q1.** ***The method relies on well aligned faces. How sensitive is the method to different face positions or image quality?***
>
> Face detection and alignment [14] is very commonly used in human face-related projects, e.g. StyleGAN [2], InsightFace [14], GPEN, and a concurrent work FastComposer [Supp. 5]. This preprocessing step is pretty low-cost, usually achieving $30+$ FPS on CPU or $200+$ FPS on GPU and having only about $10{\rm{M}}$ parameters. Besides, in many practical scenarios, only facial parts of a user can be collected, or even only a single face image. Our method requiring only one face image for each identity is acceptable for real-world application.
>
> Besides, as stated in **GQ1** and **GQ2**, `Rebuttal PDF Figure 1` shows the robustness of our method to different head poses, ages, light conditions, etc. Even given a profile face, our method can generate its frontal face or other head poses just by inputing the corresponding prompts. The robustness and good editability come from the identity disentanglement of our method. In specific, the face recognition encoder [14] pre-trained on large face datasets is robust to gender/age/pose/expression/light/etc. variants of input faces. And the celeb basis space gives good text-driven controllability to the learned representations.
>
> **R4Q2.** ***A large amount inference images are also from the celebrity dataset. Please provide more results on more personalized identities, like babies.***
>
> First, we gracefully point out that *"inference images also from the celebrity dataset"* is incorrect. To avoid the model directly re-using the celeb embeddings, we use StyleGAN [2] synthesized images (noted as **fake** images) as the personalizing targets to compare our method with other methods. These **fake** input identities are irrelevant to the celebrities, which can be considered as a nobody or ordinary person that Stable Diffusion [1] has NOT seen before.
>
> Second, our `Rebuttal PDF Figure 1` additionally shows the experimental results on babies, teenagers, and elders.
>
> **R4Q3.** ***No ablation study on #images per person.***
>
> As detailed in **GQ3**, premised on the fact our method achieves one-shot identity personalization, increasing the number of images per person $N$ from $1$ to $20$ can slightly improve the visual quality of synthesized images. Additionally, the quantitative metrics improve marginally when increasing $N$. The more comprehensive ablation study about using different $N$ will be added to the final version of our paper.
>
> **R4Q4.** ***How to generalize to other non-face samples?***
>
> The above global response **GQ4** has answered the question. With a simple replacement, we succeed in making our method extended to the "car" and "cat" personalizing tasks. Inspired by this, we'll continue studying on a more general "basis"-based method in future work.

---

> > ### Comment · Reviewer_Qr1K · 2023-08-11
> >
> > Thanks for providing very comprehensive responses, it answers most of my questions.
> > I would love to increase my score to weak accept.

---

### Official Review · Reviewer_MXrx · 2023-07-07

**Soundness:** 3 good
**Presentation:** 3 good
**Contribution:** 3 good
**Rating:** 7
**Confidence:** 5

**Summary:**

This paper introduces an innovative method for generating highly personalized images utilizing a single reference image and distinct textual prompts.The central idea revolves around identifying a subspace within the text encoder embedding space, which, when combined, can represent any facial images. This is accomplished by applying PCA to the CLIP text embedding of hundreds of celebrity names. Subsequently, a single MLP is optimized to compute the combination weights using the features of a new individual, leveraging the diffusion reconstruction loss. The proposed approach's effectiveness is demonstrated through the results on both single and multiple subject generation tasks. The method significantly outperforms previous baselines in user studies, delivering superior qualitative results with notably enhanced diversity




**Strengths:**

S1: The paper's innovative approach to identifying a text embedding subspace stands out. This subspace allows for an effective projection of subject-specific facial data while remaining compact, thereby avoiding overfitting to the reference subject.

S2: The experimental sections of the paper are comprehensive, providing detailed comparisons via quantitative, qualitative, and user studies, along with insightful ablation studies.

S3: The proposed method achieves a good balance between identity preservation and prompt consistency, outperforming popular baselines such as textual inversion and DreamBooth.





**Weaknesses:**

W1: The authors compute the textual inversion subspaces via PCA on textual embedding of celebrity names. However, celebrities might not be a representative sample of ordinary people and could inherently carry biases related to racial and geographical factors. This might bias the subsequent image generation process.

W2: The method employs subject inversion by optimizing weights in the celeb-basis. Nevertheless, the diffusion model remains static, which may result in generated samples occasionally deviating from the reference subject.

W3: The method is still optimization based, and requires costly back propagation, which might be infeasible as the diffusion model gets larger and larger.

**Questions:**

Despite the shortcomings identified above, I appreciate the overall simplicity and effectiveness of the proposed approach. Consequently, I am inclined to recommend acceptance of the paper.

**Limitations:**

See weakness sections.

---

> ### Author Rebuttal · Authors · 2023-08-10
>
> # 3. Responses to R3 (MXrx)
>
> Thank you for the careful reading and useful advice. We respond to your questions below:
>
> **R3Q1.** ***Using celebrities to represent ordinary people may bring bias?***
>
> Our global response to **GQ1** explains why limited celebrities can represent any ordinary person. Also as shown in `Rebuttal PDF Figure 1`, our method is robust to the age, head pose, gender, and light conditions with little bias.
>
> **R3Q2.** ***Freezing the diffusion model leads to the deviation?***
>
>
> Finetuning the diffusion model potentially brings the forgetting problem [10,8]: some important concepts may be "forgotten" after modifying the diffusion model weights. In our method, due to the good disentangling ability of the face recognition encoder [14] (pre-trained on large-scale face recognition datasets) and powerful priors contained in celeb basis (corresponding to over $700{\rm{K}}$ samples as shown in `Rebuttal PDF Figure 2`), we believe that learned features have strong ability to precisely represent any given person and generate text/identity-aligned images. The additional results in `Rebuttal PDF Figure 1` and uncurated results in `Rebuttal PDF Figure 3` further demonstrate this.
>
>
> **R3Q3.** ***The optimization-based method is infeasible as the diffusion model getting larger.***
>
> We agree with this opinion. However, this is indeed a common limitation of most personalization methods as far as we know. Very similar to FC [Supp. 5] which pre-trains a unified human identity feature extractor on text-image paired dataset within batch size 128 and 150k steps on 8 NVIDIA A6000 GPUs, we have also tried to train a model on 50k identities (20 faces per identity without text labels) as an encoder-based method (inference-only). However, it is hard to converge in a short GPU time within a batch size of 8. We think it might be because we need much more data and computing resources to train a unified face representation and we will explore it in the future.
>
> Apart from this, we would also like to highlight that our method is still state-of-the-art regarding two types of personalization methods (encoder-based and optimization-based). For encoder-based methods (e.g. FC [Supp. 5]), we utilize the pre-trained knowledge of Stable Diffusion [1] (i.e. celebrity names) as priors, which is still more efficient than training a domain encoder. As for the optimization-based methods (e.g. TI [7], DB [10], CD [8]), our method requires quite low computing cost ($1{\rm{K}}$ learnable parameters under $3$ minutes) and can be easily extended to a new class (see **GQ4**).

---

> > ### Comment · Reviewer_MXrx · 2023-08-17
> > **Response**
> >
> > Thank you for the rebuttal. Regarding points 1 and 2, celebrities represent a minor fraction of the overall population. I would recommend that the authors address this limitation in the finalized version. For point 3, I concur that integrating an encoder-based approach with the celeb-based concepts presented in this paper poses certain challenges, which can be explored in future works. I suggest the authors to add some discussion about this and add quantitative comparisons (image quality / efficiency) with encoder-based methods like ELITE and FastComposer as other reviewers suggest.
> >
> > On the whole, I remain optimistic about the submission and keep my original rating.

---

### Official Review · Reviewer_zJJ7 · 2023-07-07

**Soundness:** 3 good
**Presentation:** 3 good
**Contribution:** 2 fair
**Rating:** 5
**Confidence:** 4

**Summary:**

The paper presents a method to invert input portrait images to latent diffusion model (LDM)'s embedding space and then use the same to create text-based identity preserving compositional portrait images. To enable image inversion, the authors create a novel active appearance model "Celeb Basis" (image-only identity PCA) from StyleGAN-2 style frontal images. Later, this "Celeb Basis", and a textual embedding are jointly used to train CLIP for identity preserving (ArcFace style loss) image synthesis. At test time, new images of the inverted target image (represented by the Celeb Basis) can be generated based on textual prompts.

The authors presents ablation studies wrt prompt and identity preservation (and detection) and compare against SOTA methods "An image is worth one word" (Textural Inversion), Dreambooth and Multi-concept customization of text-to-image diffusion (Custom diffusion) and achieve good results for prompt-based facial images.

Note, that while other methods are not specific for portrait images, the current Celeb Basis has been built for portrait style images only.

**Strengths:**

While the idea of creating an active appearance model, Cootes et al. (Celeb basis) is quite established in the vision community; this paper attempts to "rig" (e.g. StyleRig, Tewari et al.) the LDM space to create new prompted facial image given target image (identity preservation only). This is the main contribution of the paper and is novel in that regard, while the idea of controlled prompt-based image synthesis via LDM has been explored before.

The celeb basis thus enables more control over image synthesis, where the semantic of "identity" preservation helps with a more _semantic_ control and results in better compositional results (two target images used together). These compositional results analysis (Fig 5, last two row) are mostly visual since I look at the overall image to argue if it looks more "believable".

The paper is generally easy to follow and extensive ablation studies justify the various modeling decisions made.

**Weaknesses:**

- The range of "Celeb basis": Since the method used PCA and uses a limited set of image to create the basis, it's limited by the choice of images, which is further limited by the choice of named celebrities in the LDM model itself. For example, it's unclear how well the model will represent certain age, pose and expression ranges.

- Table 1 shows that while the users prefer the author generated images on Identity and Detect metics, Textual Inversion outperforms them, while that method is not specific to facial images as such. The reason argued is that Textual Inversion outfits. Given that Textual Inversion is a generic method, while the current method is meant for facial images, this result is surprising.

- Comparison against FastComposer: Since most results presented against a concurrent work "FastComposer" seem cherry picked, the presented comparison seems biased. For example, the result presented in Fig 1 (FastComposer) seem better than those presented in Supp (Figure 5). Moreover, the current method does not explicitly model expression control in their Celeb basis, and their Fig 6 comparison seems less principled.

**Questions:**

- The data augmentation technique is essentially used color space augmentations. Did the authors consider any geometric augmentations? Since, the image inversion techniques in the StyleGAN space are quite mature (eg HyperStyle), adding augs that change the pose or expression would make the method further robust?
- How were the results shown in the paper selected? Was it the first image that the method created used or several prompt tries with subtle textual changes made to pick the results shown in the paper?
- Did the authors consider the use of multi-view images for identity reconstruction?

**Limitations:**

Novelty: The main novelty of the approach is introduction of celeb-basis, while the rest of the method seems straight forward. Since the results are only shown for a facial model, it scope limits the contribution of the work. Please rename the work from "Inserting Anybody" to "Inserting Anybody's Portrait". As such control over bodies, etc. is not part of the paper's contribution.

Expression control: While the basis does not model expressions in the 3DMM sense or provides a basis for expressions, the method seems to provide control expression control via prompts (supp. comparison against FastComposer). This seems to be an after thought and it's unclear how well the expressions are enabled by the method. In order to fully understand the expression control capabilities, please use a ground truth dataset (such as https://www.cs.cmu.edu/afs/cs/project/PIE/MultiPie/Multi-Pie/Home.html) and provide comparison against ground truth expression vs those generated by prompts. This would also help understand how well the expressions can be re-created.

Overall, I think this is an interesting paper and the results are generally good; the paper's novelty is limited since it enables only identity inversion, and later dependent on CLIP/LDM's capabilities to create pose, expression or semantic variations (e.g. an older V_1), etc. So, there's merit in the contribution. Besides the points/questions mentioned earlier, the authors should expand the limitations, move FastComposer results to the main paper.

---

> ### Author Rebuttal · Authors · 2023-08-10
>
> # 2. Responses to R2 (zJJ7)
>
> We thank you for your careful reading and helpful suggestions. We respond to your concerns below:
>
> **R2W1.** ***Using limited celeb names to represent a person may constrain the ability to control age, pose, and expression.***
>
> Please see our global response to **GQ1** and **GQ2**, where we have answered in detail.
>
> **R2W2.** ***Textual Inversion (TI) [7] results in the `main paper Table 1` are surprising.***
>
> As shown in `Figure 5 of the main paper`, TI [7] can only generate images similar to the training image and ignore the additional text prompt. Thus, TI [7] performs better on identity and detection metrics since the output image is very similar to the input one (large detailed face), which is easy to be identified. However, we argue that text-image alignment is critical for our text-driven image synthesizing task. As shown in `our main paper Table 1 and Figure 5`, the synthesizing results of TI [7] are often very similar to the input images, hardly controllable by the input text prompts. The detailed quantitative editing ability comparison in `our main paper Figure 6` also demonstrates this phenomenon.
>
> **R2W3.** ***The comparison against FastComposer [Supp. 5] seems cherry picked and less princinpled.***
>
> We follow a fair testing protocol (detailed in the later response to **R2Q2**) to compare all the methods.  `Supplement Figure 4-7` compares our method and FastComposer (FC)'s [Supp. 5] the ability to control a single person and compose multiple persons. Concretely in the experiments, for each text input, we run the FC [Supp. 5] online demo eight times and select the best one (with the highest text-image alignment score) as the compared result.
>
> During experiments, we found that FC [Supp. 5] has three main weaknesses:
>
> 1. hard to generate full body;
> 2. hard to make personalized identities interact with each other;
> 3. over-consistent facial expression and head pose.
>
> Apparently, the results in `Supplement Figure 4-7` demonstrate that our celeb basis outperforms FC [Supp. 5] on these aspects.
>
> **R2Q1.** ***What if adding augs that change the pose or expression?***
>
> These augmentations may distort the original identity.
>
> For StyleGAN [2] synthesized faces as input, in our practice, shifting the latent vectors of StyleGAN [2] may change the pose and expression of a face, but also unavoidably brings identity distortion, leading to worse results. Besides, using other face editing methods might improve the performance, but brings massive training overhead. Consequently, without these complex augmentations, our method performs well even using a very simple augmentation scheme (flip, resize, and color jitter).
>
> For realistic faces, instead of introducing other face editing methods to augment a single image, directly increasing the number of images per person seems to be a better choice, which has been detailed in **GQ3**.
>
> **R2Q2.** ***How were the results shown in the paper selected?***
>
> We pick the best 1-3 images having the highest text-alignment scores from a single inference batch (with batch size 8, and differently, we use batch size 6 in the `Rebuttal PDF` limited by the NIPS rebuttal due). This testing protocol is used for all the methods including our celeb basis and all the compared methods. The testing protocol can be vised as a peak performance comparison and we will update this information in the revised paper.
>
> We also list the uncurated samples synthesized through our method in `Rebuttal PDF Figure 3`.
>
> **R2Q3.** ***What if using multi-view images for identity reconstruction?***
>
> Please see our response to **GQ3**, where we provide the results.
>
> **R2Q4.** ***Rename the title to "Inserting Anybody's Portraits".***
>
> We thank you for your advice and will take it into account to rename a better title for our work. Note that although our method only takes as input face images, the synthesized results contain not only the identity-consistent faces but also half/full bodies, different actions/motions, variant age/pose/expression/gender, and multiple persons' interaction, all driven by text prompts. This can be considered that we successfully insert a given identity (with only a face) into the "identity" space of the diffusion model. The newly inserted identity shares the same good editability with those recognized persons (celebrities), as mentioned in **GQ1**.
>
> **R2Q5.** ***Please provide more detailed expression controlling validation results, e.g. based on the Multi-Pie groundtruth expression dataset?***
>
> We sincerely appreciate your advice. The later response to **R2Q6** will answer to this question in detail.
>
> **R2Q6.** ***Expand the limitations and move FastComposer [Supp. 5] to the main paper.***
>
> Thanks for your suggestion. You mentioned that our method has two main limitations:
> * (L1) lacking quantitative metrics above expression controlling;
> * (L2) only supporting identity inversion.
>
> For (L1), we believe the text alignment metrics already evaluate the quantitative performance of text-to-image methods. And as a supplement to this, we'll use Face Age/Expression/Pose feature extractors to further validate the controlling ability for human identity personalized tasks in the future.
>
> For (L2), please see our global response to **GQ4**, where we extend our method to "car" and "cat" classes. We'll take this insight as future work.
>
> We'll add more detailed limitations of our method and more complement comparisons with the concurrent work FC [Supp. 5] in the final version.

---

> > ### Comment · Reviewer_zJJ7 · 2023-08-17
> >
> > Thank you for the rebuttal. I am generally happy with the author replies, and keep my "acceptance" rating.

---

### Official Review · Reviewer_ABqC · 2023-07-11

**Soundness:** 3 good
**Presentation:** 3 good
**Contribution:** 4 excellent
**Rating:** 7
**Confidence:** 2

**Summary:**

The paper proposes a new method to personalize the pre-trained text-to-image model on humans. The propose a novel idea called Celeb basis and it simplifies the problem to reconstructing the coefficients of any new identity. Both quantitative and qualitative results demonstrate the effectiveness of the proposed method on concept composition compared with the baselines.

**Strengths:**

+ It's a very smart idea to utilize Celeb Basis from the text embedding space of the celebrities’ names in the text-to-image model. With the low dimension subspace, the learning problem boils down to very few coefficients, which is simple yet effective.

+ The experimental results speak for the paper itself. It shows more stable concept composition abilities than previous works.

I especially like fig 5 where the motivation of the proposed method is clearly demonstrated and easy to see where the previous methods fall short on.


**Weaknesses:**

N/A

**Questions:**

will the codebase be publicly available and when?

**Limitations:**

I think authors have adequately discussed the limitations in terms of the artifacts of the real human faces of the original stable diffusion model and other species. Potential negative impact has also been discussed.

---

> ### Author Rebuttal · Authors · 2023-08-10
>
> # 1. Responses to R1 (ABqC)
>
> **R1Q1.** ***Will the codebase be publicly available and when?***
>
> We thank you for your advice. Our codes (including the whole training and testing process) will be publicly available on GitHub as soon as the NIPS notifications are released. We are making the code easier to be used. A project webpage, a GitHub repository, a Google Colab demo, and a HuggingFace workspace will be all publicly available. Additionally, due to our method can generate features compatible with Textual Inversion [7], we will also work on developing a celeb basis as a plug-in extension of commonly used Stable Diffusion [1] WebUI.
>
> We sincerely expect your future experiences with these resources. And please feel free to give us feedback and advice if you have any problems using these resources.

---

> > ### Comment · Area_Chair_y5r5 · 2023-08-19
> >
> > Thanks to the authors for your response.
> >
> > Best regards,
> > Your AC

---

### Author Rebuttal · Authors · 2023-08-10

# 0. Global Response

We thank all reviewers for their valuable feedback and recognition of the proposed Celeb Basis in terms of the idea novelty, experiment results, and paper written. Below, we address their common concerns.

**GQ1.** ***Can limited celeb names represent any person? (asked by R2,R3,R5)***

Yes, our method uses PCA to find a compact space and fits any given person onto this space through a denoising reconstruction loss [23]. Note that Stable Diffusion [1] pretrained on a very large image-text paired dataset (LAION-5B) including thousands of samples of our used celeb names (see `Figure 2 in our Rebuttal PDF`). Once converged, the learned representation as a linear combination of celeb basis merits good text-editability. In our paper, the results on StyleGAN generated **fake** faces input demonstrate the strong representing ability. Additionally, `Figure 1 in our Rebuttal PDF` shows the robustness of our method against age, pose, gender, light, etc.

**GQ2.** ***The ability of controlling age, pose, expression, etc.? (asked by R2, R4, R5)***

The results in `Supplement Figure 6` along with `Rebuttal PDF Figure 1` show the editability of expression, age, pose, and gender, all driven by text prompts. Thanks to the identity disentangling ability of the face recognition (FR) [14] network, the learned representations are highly relevant to the input identities while irrelevant and insensitive to other attributes like age, pose, expression, etc. Furthermore, the learned representation as an interpolation of celeb basis inherits the good characteristic of those "recognized" celeb names, which corresponds to large amount of image-text paired samples used for pretraining Stable Diffusion [1].

**GQ3.** ***What if training with more images per identity? (asked by R2,R4)***

Our method focuses on one-shot identity personalization problem (i.e. training from a single photo to represent a new concept). To verify the influence of training images count $N$ per identity, we provide the results in `Figure 4 of Rebuttal PDF`. It shows that when we use more images, the face quality can also be slightly improved, and the synthesized results are already pretty good when $N=1$. We'll add these results and analyses to the final version.

**GQ4.** ***How to extend to more general classes, like car or cat? (asked by R2,R4)***

Thanks for your suggestion! We also have considered it as a limitation and future work in the `main paper Section 4.4`. Here, we provide some rough experiments. As shown in `Rebuttal PDF Figure 5`, for non-face samples, we trivially replace the FR encoder with a pre-trained ResNet50 (trained on ImageNet-1K) and construct the "car" (or "cat") basis by collecting $\sim100$ typical types of cars (or breeds of cats). Without other bells and whistles, our method generates text-aligned images with good quality of these classes. In the future, we'll try a better general encoder (to replace ResNet50) and crawl a larger basis for non-face classes, extending our method for more general subjects by different well-collected basis names.

**We also provide a `rebuttal PDF` as attached below in this global response part to show more results.**

---

### Decision · Program_Chairs · 2023-09-21

**Decision:**

Accept (poster)

**Comment:**

The paper received good feedback from the reviewers. Overall, the proposed method is smart, simple, and effective.

The ACs agree to accept the paper to NeurIPS. The authors should consider the discussions during the rebuttal phase in the camera-ready version.